# Learning Discrete Diffusion on Graphs via Free-Energy Gradient Flows

**Dario Rancati** [1]  **Jan Maas** [1]  **Francesco Locatello** [1]

## Abstract

Diffusion-based models on continuous spaces have seen substantial recent progress through the mathematical framework of gradient flows, leveraging the Wasserstein-2 ($W_2$) metric via the Jordan-Kinderlehrer-Otto (JKO) scheme. Despite the increasing popularity of diffusion models on discrete spaces using continuous-time Markov chains, a parallel theoretical framework based on gradient flows has remained elusive due to intrinsic challenges in translating the $W_2$ distance directly into these settings. In this work, we propose the first computational approach addressing these challenges, leveraging an appropriate metric $W_K$ on the simplex of probability distributions, which enables us to interpret widely used discrete diffusion paths, such as the discrete heat equation, as gradient flows of specific free-energy functionals. Through this theoretical insight, we introduce a novel methodology for learning diffusion dynamics over discrete spaces, which recovers the underlying functional directly by leveraging first-order optimality conditions for the JKO scheme. The resulting method optimizes a simple quadratic loss, trains extremely fast, does not require individual sample trajectories, and only needs a numerical preprocessing computing $W_K$-geodesics. We validate our method through extensive numerical experiments on synthetic data, showing that we can recover the underlying functional for a variety of graph classes, as well as cell type prediction in single-cell trajectory data. Code will be made available at  github.com/dariorancati/discrete-gradient-flow.

[1]Institute of Science and Technology Austria. Correspondence to: Dario Rancati <FirstName.LastName@ist.ac.at>.

*Proceedings of the 43$^{rd}$ International Conference on Machine Learning*, Seoul, South Korea. PMLR 306, 2026. Copyright 2026 by the author(s).

## 1. Overview

Over the last few years, a broad family of diffusion-based methods has modeled the evolution of a particle population as a trajectory of probability measures in a metric space (Bunne et al., 2022; Terpin et al., 2024; Xu et al., 2024a; Choi et al., 2024; Mokrov et al., 2021; Salim et al., 2021; Haviv et al., 2025). A central tool behind many of these approaches is the following discretization scheme, introduced by Jordan, Kinderlehrer and Otto in their seminal paper (1998):

$$p_{t+1} := \arg\min_{p \in \mathcal{P}(\mathbb{R}^n)} \left\{ \mathcal{F}(p) + \frac{1}{2\tau} W_2(p, p_t)^2 \right\}. \quad (1)$$

Each step of this scheme (typically called the JKO scheme) solves a proximal optimization problem in the Wasserstein-2 ($W_2$) geometry under the influence of the functional $\mathcal{F}$. This line of work is mostly known for the striking mathematical property of identifying certain PDEs, such as the Fokker-Planck equation, with the gradient flow (Ambrosio et al., 2008) of free-energy functionals which are the sum of a potential and an entropic term. Nonetheless, this variational viewpoint is also attractive from the practical point of view of machine learning: models built off this idea typically exhibit fast training loops and operate directly at the sample level, with no need for trajectory-level observations (Bunne et al., 2022; Santambrogio, 2015).

In parallel, models based on probability flows on discrete spaces have recently gained significant popularity, impacting areas such as language modelling, molecule generation and machine learning for material sciences (Benton et al., 2024; Lou et al., 2024; Campbell et al., 2022; Shi et al., 2023; Campbell et al., 2024; Stark et al., 2024; Davis et al., 2024; Xu et al., 2024b; Kim et al., 2025). These models typically study discrete probability evolutions structured by an underlying, known, Markov kernel $K$ that specifies the state-transition probabilities at each time-step. In this context, however, a direct translation of the JKO framework in the $W_2$ metric breaks down: for finite state spaces, any non-constant curve of probability distributions has infinite $W_2$-metric derivative, hence common trajectories such as the discrete heat equation cannot be interpreted as a $W_2$-gradient flow of *any* functional.

In this work, we introduce the first machine learning method that addresses this difficulty, allowing us to learn functionals from their gradient-flow trajectories in the space of probability measures via a JKO-style proximal scheme. Our framework builds on a body of mathematical works on probability gradient flows over discrete spaces (Maas, 2011; Mielke, 2011; Chow et al., 2012; Erbar & Maas, 2012a;b; Erbar et al., 2017; 2018; Chow et al., 2017). These works rely on the use of a different metric $W_K$ depending on the Markov kernel $K$ through a discrete analog of the Benamou-Brenier formula (Benamou & Brenier, 2000). We stress that these constructions were developed in pure mathematics, and it is not obvious that they would have computational implications. In fact, they have seen almost no practical adoption, with minor exceptions in applied mathematics (Erbar et al., 2017), and, within machine learning, only in the specific case of discrete samplers (Sun et al., 2023). In our work, we translate this theory to derive a practical, implementable methodology, building a principled discrete analogue of the JKO scheme, which is already successful in continuous settings.

Within the geometry induced by $W_K$, we use the fact that important discrete diffusion trajectories (such as the discrete heat equation on a graph) arise as gradient flows of free-energy functionals, in direct analogy with the continuous case. We devise a strategy to learn these functionals given this characterization, based on first-order necessary conditions of optimality for our discrete JKO scheme, requiring only the computation of geodesics in $W_K$ as a preprocessing step, for which we build a customized and efficient numerical routine. The resulting training loop is lightweight, and in experiments we benchmark extensively and outperform standard Markov jump-process baselines.

We summarize our contributions as follows:

- We translate the theoretical framework for probability gradient flows in discrete spaces into the context of machine learning. In particular, we focus on the tools that enable applications of this theory to computational domains. While this theory already existed in pure mathematics, it had seen close to no computational applications and was largely inaccessible to researchers in the ML field. A key barrier was that results on the uniqueness of solutions for the discrete JKO scheme and on the form of its gradients in the $W_K$ metric are critical for implementation, but were missing.

- Based on said framework, we devise a practical strategy to learn discrete gradient flows from temporal snapshots. Our approach is based on imposing first-order necessary conditions for the discrete JKO scheme, and its main components are a numerical routine to numerically compute geodesics and a quadratic-loss training

loop based on said geodesics;

- We extensively benchmark our framework on a variety of graph classes, sizes and regimes. Through comparison with existing methods for learning Markov Jump Process dynamics, we find our method to have better performance while being extremely favorable in both training time and number of parameters. We also verify its effectiveness on real, single-cell trajectory data.

## 2. Problem Setup

We are interested in studying probability flows on a finite space $\mathcal{X} = \{x_1, ..., x_N\}$ equipped with a Markov transition kernel $K \in \mathbb{R}^{N \times N}$, that is, a non-negative, row-stochastic real matrix. We make the standard assumption for $K$ to be irreducible and reversible (Campbell et al., 2022; Lou et al., 2024): irreducibility implies in particular the existence and uniqueness of an invariant probability measure $\pi$ such that $\pi^T K = \pi^T$. We denote by:

$$\mathcal{P}(\mathcal{X}) :=$$
$$\left\{ \rho : \mathcal{X} \to \mathbb{R}, \ \ \rho(x) \geq 0 \ \ \forall x \in \mathcal{X}, \ \ \sum_{x \in \mathcal{X}} \pi(x)\rho(x) = 1 \right\}$$

the set of all *probability densities* on $\mathcal{X}$, and by $\mathcal{P}_*(\mathcal{X}) \subseteq \mathcal{P}(\mathcal{X})$ the subset of densities that are strictly positive. Throughout the paper, we use $\rho$ to denote densities and $p$ to denote probabilities. The core example of the probability evolution processes we are interested in is the *discrete heat equation*:

$$\frac{\mathrm{d}}{\mathrm{d}t}\rho_t = (K - I)\rho_t, \tag{2}$$

which is the most widely used process in discrete generative applications (Lou et al., 2024; Campbell et al., 2022). Our goal is to describe processes such as (2) as gradient flows of functionals on $\mathcal{P}(\mathcal{X})$, mirroring the continuous case (Jordan et al., 1998; Bunne et al., 2022; Terpin et al., 2024): in these works, the continuous counterparts to (2) (diffusion SDEs of the kind $\mathrm{d}X_t = f(X_t, t)\mathrm{d}t + \sigma(t)\mathrm{d}B_t$) are learned by leveraging the JKO scheme (1) to describe the evolution of their density $p_t \sim X_t$. This allows them to cast the problem of learning $p_t$ as that of learning $\mathcal{F}$ from temporal snapshots $\{X_t\}_{t \in [t_0, t_1, \cdots, t_M]}$. The functional $\mathcal{F}$ is usually constrained to have a free-energy form, i.e. the sum of a potential and entropy term $\mathcal{V} + \beta\mathcal{H}$, which the JKO scheme (Jordan et al., 1998) ensures can describe diffusion SDEs.

However, it turns out that the Wasserstein-2 metric that enables the JKO approach on continuous domains fails to achieve the same on any discrete space, even in the simple $N = 2$ setting.

**Lemma 2.1** (Informal). *Let $\mathcal{X} := \{a, b\}$ be a 2-point space, and let $K$ be a Markov kernel on $\mathcal{X}$ with invariant probability measure $\pi$. Let $\rho \neq \mathbf{1}$ be any non-invariant density on $\mathcal{X}$, and let $\rho_t$ denote the solution of the discrete heat equation (2) with $\rho_0 = \rho$. Define $p_t = \rho_t \odot \pi$ to be the probability $p_t$ associated to the density $\rho_t$. Then, for all $t \geq 0$:*

$$|\dot{p}_t| := \limsup_{s \to t} \frac{W_2(p_t, p_s)}{|t - s|} = +\infty.$$

For a formal version with proof of this lemma, see Appendix C. As a consequence of Lemma 2.1, the heat equation cannot be interpreted as the gradient flow of *any* functional on $\mathcal{P}(\mathcal{X})$ equipped with the standard $W_2$ metric. If one hopes to derive a similar approach in the discrete case which is at least descriptive enough to include processes such as (2), one inevitably needs to look at different metrics than $W_2$.

## 3. Revisiting the Theory of Discrete Gradient Flows

In this section, we introduce the geometry that will enable our discrete JKO approach. Here we provide the minimal results needed for our application, and we refer the reader to Appendix B for more details. We emphasize again that, while the results in this section are known to the discrete optimal transport community, to our knowledge a concise summary aimed at a machine learning audience is missing.

### 3.1. The metric $W_K$

The key idea to build an alternative metric is to use a discrete parallel of the dynamic formulation of optimal transport due to Benamou & Brenier (2000). To do this, one needs to define an appropriate *mobility* on the graph associated to the kernel $K$, which models how the cost of moving mass from $x_i$ to $x_j$ depends on $\rho(x_i)$ and $\rho(x_j)$. To describe the discrete heat equation (2) as gradient flow of the entropy, it turns out (Maas, 2011) that the appropriate choice is the *logarithmic mean* $m_\rho(x_i, x_j)$ given by:

$$m_\rho(x_i, x_j) := \int_0^1 \rho(x_i)^s \rho(x_j)^{1-s} \mathrm{d}s =$$
$$= \begin{cases} \frac{\rho(x_i) - \rho(x_j)}{\log \rho(x_i) - \log \rho(x_j)} & \rho(x_i) \neq \rho(x_j) \\ \rho(x_i) & \rho(x_i) = \rho(x_j) \end{cases}. \quad (3)$$

Other possible choices are the geometric mean $\sqrt{\rho(x_i)\rho(x_j)}$ and, more generally, $\rho(x_i)^\alpha \rho(x_j)^\alpha$ for $\alpha > 0$. We refer the reader to Maas (2011); Erbar & Maas (2012b); Erbar et al. (2018) for a study of dynamical transport metrics with general mobilities. The choice of the mobility function is a modelling choice that directly relates to the kind of random dynamics one expects to observe.

Here we focus on the logarithmic mean because of the importance of the discrete heat equation (2). If the expected noise process is different, one can model it by tuning the mobility: for example, this has been done for the discrete porous medium equation (Erbar & Maas, 2012b).

With this insight, we are ready to define the alternative metric $W_K$ that will be leveraged in this paper:

**Definition 3.1** (Metric $W_K$). For $\rho_0, \rho_1 \in \mathcal{P}(\mathcal{X})$ we define:

$$W_K(\rho_0, \rho_1)^2 := \quad (4)$$
$$\inf_{\rho, \psi} \left\{ \frac{1}{2} \int_0^1 \sum_{x, y \in \mathcal{X}} (\psi_t(x) - \psi_t(y))^2 K(x, y) m_{\rho_t}(x, y) \pi(x) \mathrm{d}t \right\}$$

where the infimum is taken over all piecewise $C^1$ curves $\rho : [0, 1] \to \mathcal{P}(\mathcal{X})$ and measurable $\psi : [0, 1] \to \mathbb{R}^{\mathcal{X}}$ satisfying at almost every time $t$ the *discrete continuity equation*:

$$\begin{cases} \dot{\rho}_t(x) + \sum_{y \in \mathcal{X}} (\psi_t(y) - \psi_t(x)) K(x, y) m_{\rho_t}(x, y) = 0 \\ \rho(0) = \rho_0, \quad \rho(1) = \rho_1 \end{cases} \quad (5)$$

Intuitively, the distance between two densities can be interpreted as the minimal "discrete kinetic energy" required to transform a density $\rho_0$ into another density $\rho_1$. (Here, $\psi(x) - \psi(y)$ is viewed as a discrete velocity vector along an edge $(x, y)$).

### 3.2. Geometry of $(\mathcal{P}(\mathcal{X}), W_K)$

Having defined the metric $W_K$, we cite some of the properties of the spaces $\mathcal{P}(\mathcal{X})$ and $\mathcal{P}_*(\mathcal{X})$ equipped with the metric $W_K$ which will be useful later. First, we denote the discrete gradient of a function $\psi \in \mathbb{R}^N$ by

$$\nabla \psi \in \mathbb{R}^{N \times N} \quad, \quad \nabla \psi(x, y) := \psi(x) - \psi(y). \quad (6)$$

We sometimes write $\nabla \psi(x)$ to indicate the vector $[\psi(x) - \psi(y)]_{y \in \mathcal{X}}$. In order to study functionals on $(\mathcal{P}_*(\mathcal{X}), W_K)$, we need to equip this space with a Riemannian manifold structure. For this purpose, it will be natural to identify the tangent space at a point $\rho \in \mathcal{P}_*(\mathcal{X})$ with the collection of discrete gradients:

$$T_\rho := \left\{ \nabla \psi \in \mathbb{R}^{N \times N} : \psi \in \mathbb{R}^N \right\}. \quad (7)$$

This can be done as it turns out that, for each fixed $t$ and fixed $\rho_t$, the discrete continuity equation (5) defines a 1-1 correspondence between the "vertical derivative" $\dot{\rho}_t$ and the "horizontal derivative" $\nabla \psi_t$; see (Maas, 2011). One can then construct a Riemannian structure on $\mathcal{P}_*(\mathcal{X})$ by equipping $T_\rho$ with the following inner product $\langle \cdot, \cdot \rangle_\rho$ (see B.6 in the appendix):

$$\langle \nabla \phi, \nabla \psi \rangle_\rho := \tag{8}$$
$$\frac{1}{2} \sum_{x,y \in \mathcal{X}} (\phi(x) - \phi(y))(\psi(x) - \psi(y)) K(x,y) m_\rho(x,y) \pi(x).$$

With this choice, we can complete the construction of the Riemannian structure that we will leverage:

**Theorem 3.2** ((Maas, 2011), Thm. 3.29). *The space $(\mathcal{P}(\mathcal{X}), W_K)$ is a complete metric space, and the space $\mathcal{P}_*(\mathcal{X})$ endowed with the scalar products* (8) *is a Riemannian manifold. The induced Riemannian distance on $\mathcal{P}_*(\mathcal{X})$ coincides with $W_K$.*

Only the space of strictly positive densities $\mathcal{P}_*(\mathcal{X})$ gets the smooth structure of a Riemannian manifold, as the Riemannian metric defined above becomes degenerate at the boundary of the probability simplex; the whole $\mathcal{P}(\mathcal{X})$ is only a metric space. This will be crucial later, as we'll need to ensure the solution of our discrete JKO scheme fall in the interior of the simplex.

Finally, we are ready to describe the discrete heat equation (2) as the gradient flow of the Kullback-Leibler divergence in the space $(\mathcal{P}_*(\mathcal{X}), W_K)$.

**Theorem 3.3** ((Maas, 2011), Thm. 4.7). *Define the Kullback-Leibler divergence w.r.t. the invariant measure $\pi$ as:*

$$\mathcal{H}(\rho) := \sum_{x \in \mathcal{X}} \rho(x) \log \rho(x) \pi(x). \tag{9}$$

*Then the discrete heat equation* (2) *coincides with the gradient flow equation for the metric $W_K$, given by $D_t \rho = -\mathrm{grad}\, \mathcal{H}(\rho_t)$.*

This result is the payoff for introducing the metric $W_K$ and using the logarithmic mean mobility: with this metric, the discrete heat equation (2) is the gradient flow of $\mathcal{H}$ just like the continuous heat equation $\mathrm{d}X_t = \mathrm{d}B_t$ is the gradient flow of the Shannon entropy $-\int p(x) \log p(x) \mathrm{d}x$ for the $W_2$ metric. We can now cast the problem of learning the dynamics of processes such as $\rho_t$ as that of learning the underlying functional $\mathcal{F}$.

## 4. Learning Discrete Gradient Flows

In the next sections we introduce most of the methodological contributions of this paper: we prove existence and uniqueness of solutions for the discrete JKO scheme, we describe gradients of the JKO functionals in the metric $W_K$, we describe a strategy to learn said functional's gradients via first-order optimality conditions and we introduce a numerical routine to compute $W_K$-geodesics that makes this computation possible.

Following the continuous setting, we assume to have access to time snapshots of the evolution of an evolving population $\mathcal{D} := \{\mathcal{Y}_{t_0=0}, \cdots, \mathcal{Y}_{t_M=T}\}$ recorded at a set of timesteps $[t_0, \cdots, t_M]$. We assume the population datasets $\mathcal{Y}_t$ to be empirical samples from a density $\rho_t$, and for this density to evolve according to the following discretized gradient flow dynamics according to an underlying, unknown functional $\mathcal{F}$ in the space $\mathcal{P}_*(\mathcal{X})$.

$$\rho_{t+1} = \arg\min_{\rho \in \mathcal{P}_*(\mathcal{X})} \left\{ \mathcal{F}(\rho) + \frac{1}{2\tau} W_K(\rho, \rho_t)^2 \right\} \tag{10}$$

Our goal is to learn $\mathcal{F}$ from the snapshots $\mathcal{Y}_t$. We present an approach based on first-order optimality conditions for (10).

### 4.1. Intuition in $\mathbb{R}^d$

We first present the intuition of the idea for the case of functionals in the space $\mathbb{R}^d$ before generalizing it to $\mathcal{P}_*(\mathcal{X})$. Let $F : \mathbb{R}^d \to \mathbb{R} \cup \{+\infty\}$ be a functional and define the iterative proximal scheme:

$$x_{t+1} = \arg\min_{x \in \mathbb{R}^d} \left\{ F(x) + \frac{1}{2\tau} \|x - x_t\|_2^2 \right\}. \tag{11}$$

If the functional $F$ is sufficiently smooth, then an easy way to extract a necessary condition for the optimality of a point $\hat{x}_{t+1}$ is to take the gradient and set it to zero:

$$\hat{x}_{t+1} \text{ is optimal for (11)} \implies \nabla F(\hat{x}_{t+1}) + \frac{\hat{x}_{t+1} - x_t}{\tau} = 0, \tag{12}$$

and the converse is true under some convexity condition on $F$. This suggests to seek the functional $F$ that, given a sequence of observations $(x_0, \cdots, x_T)$, minimizes the objective:

$$\min_F \sum_{t=0}^{T-1} \left\| \nabla F(\hat{x}_{t+1}) + \frac{\hat{x}_{t+1} - x_t}{\tau} \right\|_2^2. \tag{13}$$

We will now see how to use the Riemannian structure to construct a similar procedure in $(\mathcal{P}_*(\mathcal{X}), W_K)$.

### 4.2. Gradients of Free-Energy functionals

For the proofs of the results in this section we refer the reader to Appendix C. We now restrict to the case where the functional $\mathcal{F}$ is a free energy function, that is the sum of an arbitrary potential and the relative entropy/KL divergence. This mirrors the continuous case, and models the dynamics as a combination of a deterministic drift (the potential $\mathcal{V}$) and a random fluctuation following the heat equation:

$$\mathcal{F}(\rho) := \mathcal{V}(\rho) + \beta \mathcal{H}(\rho) = \qquad (14)$$

$$= \sum_{x \in \mathcal{X}} V(x)\rho(x)\pi(x) + \beta \sum_{x \in \mathcal{X}} \rho(x)\log\rho(x)\pi(x),$$

where $\{V(x) \in \mathbb{R}\}_{x \in \mathcal{X}}$ and $\beta \in \mathbb{R}_+$ are the unknown parameters of this functional. The following result guarantees the existence of solutions and gives us a "gradient equal to zero" condition equivalent to the Euclidean one.

**Theorem 4.1.** *Consider the discrete JKO scheme* (10) *for the discrete free energy functional:*

$$\rho_{t+1} = \underset{\rho \in \mathcal{P}_*(\mathcal{X})}{\arg\min} \left\{ \mathcal{F}(\rho) + \frac{1}{2\tau} W_K(\rho, \rho_t)^2 \right\} \qquad (15)$$

*with* $\mathcal{F}(\rho) := \mathcal{V}(\rho) + \beta\mathcal{H}(\rho)$ *with* $\beta > 0$. *Then for any* $\tau > 0$ *this scheme has a unique minimizer* $\hat{\rho}_{t+1}$ *in* $\mathcal{P}_*(\mathcal{X})$. *If* $\hat{\rho}_{t+1}$ *is this minimizer, let* $\mathrm{grad}$ *be the gradient operator at the point* $\hat{\rho}_{t+1}$ *w.r.t. the* $W_K$*-induced metric. Then:*

$$\mathrm{grad}\left(\mathcal{F}(\hat{\rho}_{t+1}) + \frac{1}{2\tau} W_K(\hat{\rho}_{t+1}, \rho_t)^2\right) = 0. \qquad (16)$$

*Remark* 4.2. The non-trivial aspect of Theorem 4.1 is the minimizer being unique and lying in the *interior* of $\mathcal{P}(\mathcal{X})$ for any $\tau > 0$, allowing us to leverage the Riemannian manifold structure. In the proof of this result, the entropy $\beta H$ crucially acts as a log-barrier at the boundary: similar results can be obtained for more general functionals relaxing the hypothesis "for any $\tau > 0$" and for example restricting to small $\tau$ values.

To construct a loss function from the first-order conditions (16), the only missing piece is to characterize the gradient of the proximal functional for the free energy $\mathcal{G}(\rho) = \mathcal{V}(\rho) + \beta\mathcal{H}(\rho) + \frac{1}{2\tau} W_K(\rho, \rho_t)^2$ for a fixed starting point $\rho_t$. We do that with the following result:

**Theorem 4.3.** *Let* $\mathcal{G}(\rho) = \mathcal{V}(\rho) + \beta\mathcal{H}(\rho) + \frac{1}{2\tau} W_K(\rho, \rho_t)^2$ *for a fixed starting point* $\rho_t$. *Then:*

$$\mathrm{grad}\,\mathcal{G}(\rho) = \nabla V + \beta\nabla\log(\rho) - \frac{1}{\tau} v_{geo}(\rho \to \rho_t), \quad (17)$$

*where* $\nabla$ *is the discrete gradient operator and* $v_{geo}(\rho \to \rho_t)$ *is the starting velocity vector in* $T_\rho$ *of the unique unit-speed geodesic starting from* $\rho$ *and passing through* $\rho_t$.

From (16) and this theorem, we can devise a strategy to learn the guiding underlying quantities $\nabla V$ and $\beta$ from the snapshots $\{\mathcal{Y}_t\}_{0 \leq t \leq T}$. The core idea is to preprocess the data by estimating empirical densities $\hat{\rho}_t$ and from there estimating the geodesic vectors at every timestep $v_t :=$

$v_{geo}(\hat{\rho}_t \to \hat{\rho}_{t-1})$. We then estimate $\nabla V$ and $\beta$ via a neural network minimizing the loss function:

$$\int_{t=0}^{T} \mathbb{E}_{x_t \sim p_t} \left[ \left\| \nabla V_\theta(x_t) + \beta_\theta \nabla \log\hat{\rho}_t(x_t) - \frac{v_t(x_t)}{\tau} \right\|_2^2 \right] \mathrm{d}t \tag{18}$$

where $p_t := \rho_t \odot \pi$ is the probability distribution computed from the density $\rho_t$ and $\rho_t(x_t)$ and $v_t(x_t)$ are defined by taking the respective vector entry corresponding to state $x_t$. We refer to Figure 4 in the Appendix for a graphical rendition of the pipeline.

*Remark* 4.4. Our loss function is based on the sole necessary condition of imposing gradients of functionals to be equal to 0. As a result, the loss function working in practice is non-obvious, as for non-convex functionals $\mathcal{F}$ there might be points in the domain that have the gradient equal to 0 while not being global minima: while the experiment results indicate that this does not happen in practice, the convexity of $\mathcal{H}$ is in general an open problem and beyond the scope of this paper. We refer the reader to (Erbar et al., 2018) for some preliminary results, which illustrate how this convexity unsurprisingly heavily depends on the graph $K$, with more "tree-like" structures being less convex than more "complete graph-like" structures. On this note, of the two most prominent $K$ used in sequence generation applications, the Hamming graph with uniform transition kernel does indeed make $\mathcal{H}$ geodesically convex, while it is to our knowledge not known whether this happens for the regularized absorbing kernel.

## 5. Numerical Estimation of $W_K$ Geodesics

We now describe how we estimate the geodesic velocity vectors $v_t$: the procedure is rather straightforward, relying on formulating the optimization problem defining $W_K$ (4) as a quadratic problem and then solving the associated KKT conditions via the Schur complement and the Cholesky factorization (Golub & van Loan, 2013), thanks to the optimality of geodesics for a Riemannian metric.

We first show how to simply reduce (4) to a quadratic optimization problem. Given $\rho \in \mathcal{P}_*(\mathcal{X})$, define the logarithmic mean matrix $\Theta(\rho) \in \mathbb{R}^{N \times N}$ by $[\Theta(\rho)]_{ij} = m_\rho(x_i, x_j)$ using (3) and also define $W(\rho) = \mathrm{diag}(\pi)\left(K \odot \Theta(\rho)\right)$. Now, define the *continuity operator* and the *graph Laplacian* as:

$$A(\rho) := K \odot \Theta(\rho) - \mathrm{diag}((K \odot \Theta(\rho))\mathbf{1}) \tag{19}$$

$$M(\rho) := \mathrm{diag}(W(\rho)\mathbf{1}) - W(\rho). \tag{20}$$

If $K$ is reversible, the matrix $M$ is symmetric (Golub & van Loan, 2013). Now, given a density derivative $\dot{\rho}$ with

$\langle \dot{\rho}, 1 \rangle_{\pi} = 0$, the optimization problem (4) becomes:

$$\min_{\psi \in \mathbb{R}^N} \quad \psi^{\top} M(\rho)\, \psi \quad \text{s.t.} \quad A(\rho)\, \psi = -\dot{\rho}, \quad \psi_p = 0,$$

where we fix a gauge by pinning one component $\psi_p$ to 0. Now, we write the KKT conditions for this optimization problem:

$$\begin{bmatrix} 2M(\rho) & A(\rho)^{\top} \\ A(\rho) & 0 \end{bmatrix} \begin{bmatrix} \psi \\ \lambda \end{bmatrix} = \begin{bmatrix} 0 \\ c \end{bmatrix}. \qquad (21)$$

From this system, it's easy to recover the solution $\psi$ as $\psi = -\frac{1}{2} M(\rho)^{-1} A(\rho)^{\top} \lambda$. We now construct $\psi$ efficiently by a double-Cholesky factorization, avoiding explicitly inverting $M(\rho)$.

1. Factor $M(\rho) = L_M L_M^{\top}$ in a lower triangular and upper triangular factors via Cholesky.

2. Solve $L_M L_M^{\top} Y = A(\rho)^{\top}$ by two triangular solves, and define $Y = M(\rho)^{-1} A(\rho)^{\top}$;

3. Assemble the Schur complement $S(\rho) = A(\rho)\, Y$;

4. Factor the Schur component, which is once again symmetric $S(\rho) = L_S L_S^{\top}$, once again via Cholesky.

5. Solve $L_S L_S^{\top} \lambda = -2c$. Finally, set $\psi = -\frac{1}{2} Y \lambda$.

This realizes the exact KKT solution using only triangular solves and two Cholesky factorizations. As the naive Cholesky factorization is cubic in the number of constraints, this routine runs in $\mathcal{O}(N^3)$ time, where we recall $N$ is the size of the set $\mathcal{X}$.

# 6. Experiments

We evaluate our method on two complementary settings: controlled synthetic discrete diffusion tasks, where the underlying graph structure and free-energy functional are known and the recovery of the learned parameters can be assessed directly, and a real-world dataset of single-cell trajectory inference in mice during the gastrulation phase of embryonic development (Pijuan-Sala et al., 2019). We complement these comparisons with targeted ablations on graph size and the number of samples.

## 6.1. Setup

We summarize here the elements shared across all evaluations: the metrics used to compare methods, the baselines we benchmark against, and the inference scheme. Unless stated otherwise, these components are kept fixed throughout the experiments.

### 6.1.1. METRICS

To evaluate the forecasting abilities of our model, we follow Berghaus et al. (2025) and use the Hellinger distance between the ground-truth probabilities and the ones estimated from our samplings. We report the time-averaged quantity $\frac{1}{T} \sum_{t=0}^{T} H(p^t, q^t)$ as $t$ varies in the considered time range, and use Hellinger as our main forecasting metric across all experimental settings.

In the synthetic setting, where the ground-truth potential $V$ and noise parameter $\beta$ are known by construction, we additionally report relative $L^2$ errors on the recovered parameters,

$$\mathcal{E}_{\nabla V} := \frac{\|\nabla V_{\theta} - \nabla V\|_2}{\|\nabla V\|_2}, \qquad \mathcal{E}_{\beta} := \frac{|\beta_{\theta} - \beta|}{|\beta|},$$

to assess whether the method recovers interpretable versions of the underlying parameters of the dynamics, rather than merely fitting the observed densities.

### 6.1.2. BASELINES

We compare our method against two baselines representative of distinct points in the space of approaches to learning discrete stochastic dynamics from snapshots.

**OpenFIM.** OpenFIM (Berghaus et al., 2025) is a recent foundation model for predicting Markov jump processes in a zero-shot setting. It is trained on a large corpus of small graphs and is designed to generalize across graph structures without task-specific retraining. As OpenFIM is restricted to the regime on which it was pretrained, we use it only in the synthetic setting and restrict the corresponding evaluation to graphs of size at most six to match its pretraining distribution.

**Direct-FP.** As a second baseline, we consider a direct optimization of the forward Kolmogorov equation associated with the free-energy functional. Given the same estimated densities $\{\hat{\rho}_t\}$ used by our method, this baseline parametrizes $\nabla V$ and $\beta$, assembles a rate matrix $Q$ associated with the free-energy functional, propagates densities forward via the matrix exponential, and minimizes the squared Hellinger distance between the propagated densities and the observed ones via Adam. Unlike our method, Direct-FP does not exploit the underlying $W_K$ geometry of the problem and operates purely in density space, optimizing the Hellinger distance directly. We apply this baseline in both the synthetic and the real-world settings.

### 6.1.3. INFERENCE

Both data generation and inference are performed using a simple Euler discretization of the forward Kolmogorov equation associated with the free-energy functional, given

the gradient $\nabla V$ and noise parameter $\beta$. We do not employ more sophisticated discrete sampling techniques, such as $\tau$-leaping (Gillespie, 2001), for simplicity, although such methods could further improve sampling efficiency. Additional details are provided in Appendix D.

## 6.2. Synthetic Experiments

### 6.2.1. MODEL

We use a deliberately simple architecture consisting of a two-hidden-layer MLP with 64 neurons per layer. The network outputs a potential head of dimension $n$, which for each basis element $x_i = e_i$ predicts the discrete gradient

$$\nabla V_\theta(x_i) = [V(y) - V(x_i)]_{y \in \mathcal{X}},$$

as well as a one-dimensional head estimating a global noise parameter $\beta_\theta$. Models are trained for two epochs with a batch size of 128 on a single L40S GPU. Training is computationally lightweight, with a single epoch taking on the order of *tens of seconds* for moderate graph sizes.

### 6.2.2. DATA

We evaluate our method on a collection of synthetic graph classes exhibiting a range of sparsity patterns and structural properties. For each class, graph instances are generated by randomly sampling both the graph topology and edge weights. Details of the graph generation procedure, along with examples of each class, are provided in Appendix H. For each graph instance, a ground-truth free-energy functional is constructed by sampling a vertex-wise potential $V$, together with a noise parameter $\beta$. An initial probability distribution $p_0$ is also sampled and used consistently for both training and evaluation. Dynamics are generated on a regular time grid with a fixed horizon, shared across training and testing.

### 6.2.3. RESULTS AGAINST BASELINES

**OpenFIM** We benchmark our method against OpenFIM (Berghaus et al., 2025), a recent foundation model for predicting Markov jump processes in a zero-shot setting. To ensure a fair and in-distribution comparison, we restrict this evaluation to graphs of size at most six, matching the regime on which OpenFIM was pretrained.

We evaluate our method across all synthetic graph classes by randomly sampling the underlying potential $V$ in the range $[-1, 1]$ and noise parameter $\beta \in [0.01, 0.1, 0.2]$ following (Terpin et al., 2024), fixing the number of samples to 10,000 per run. For each configuration, we generate five independent graph instances and report the average Hellinger distance. Results are summarized in Table 2 and Figure 1. Across all graph classes and noise levels, our method consistently outperforms OpenFIM with a clear margin. Notably,

this performance is achieved using a substantially simpler architecture, with significantly fewer parameters (for $N = 6$, typically of the order of 10k parameters) and dramatically reduced training time (under one minute, compared to several hours of pretraining for OpenFIM). While OpenFIM is designed for zero-shot prediction and trained in a different regime, we view these results as strong evidence that explicitly leveraging the underlying gradient-flow structure provides an effective and lightweight approach to learning discrete stochastic dynamics.

**Direct-FP** We next compare our method against the Direct-FP baseline introduced in Section 6.1.2. Since this baseline does not rely on a pretrained model and is not restricted to small graph sizes, we run this comparison at a larger scale than the OpenFIM benchmark: we evaluate one instance for each graph class with graph size $N = 30$ and 5,000 samples per snapshot. Both methods receive the same estimated empirical densities, so that any difference in performance reflects the value of the underlying geometric prior rather than discrepancies in the available information. We report Hellinger distance together with the relative $L^2$ errors on $\nabla V$ and $\beta$ introduced in Section 6.1.1, aggregated over all graph classes.

*Table 1.* Comparison against the Direct-FP baseline, aggregated over all graph classes at $N = 30$ and 5,000 samples per snapshot.

| Method | Hellinger | $\mathcal{E}_{\nabla V}$ | $\mathcal{E}_\beta$ |
|---|---|---|---|
| Ours | **0.152** | **0.103** | **0.376** |
| Direct-FP | 0.182 | 1.873 | 1.008 |

Results are summarized in Table 1. Despite our method not optimizing Hellinger directly, it is slightly better than Direct-FP on this metric and significantly better at recovering the underlying parameters of the dynamics. The direct baseline appears to learn a spurious solution that fits Hellinger reasonably well but fails to identify the ground-truth $\nabla V$ and $\beta$. The discrepancy between the errors on $\nabla V$ and $\beta$ is also reasonable: with $\beta$ sampled in the $[0.01, 0.2]$ range, the effect of $\nabla V$ on the dynamics is much stronger than that of $\beta$, making $\beta$ harder to identify from snapshots alone. We view this comparison as direct evidence that the $W_K$ geometry provides value beyond naive density matching, allowing the recovery of interpretable parameters of the underlying dynamics.

## 6.3. Single-cell Experiments

### 6.3.1. DATASET AND TASK

We consider single-cell trajectory inference, a prominent application domain for optimal-transport-based methods (Tong et al., 2023; Bunne et al., 2022; Terpin et al., 2024). We use the mouse gastrulation atlas of (Pijuan-Sala et al.,

2019), a standard benchmark in the trajectory inference literature, which contains approximately 116,000 cells distributed across 8 developmental timepoints. Each cell is associated with a high-dimensional gene-expression profile, an expert-annotated label from a set of 34 cell types, and a known developmental lineage graph encoding the admissible transitions between cell types (Klein et al., 2023; Weiler et al., 2023).

In principle, one would like to model these dynamics directly at the level of gene-expression states. The size of the associated discrete state space, however, places this regime outside the scope of the present framework: typical scRNA-seq datasets contain on the order of ∼50,000 genes. Following standard practice in the trajectory inference literature, we therefore work at the level of expert-annotated cell types, which can be naturally accommodated by our setup. We construct the discrete state space $\mathcal{X}$ from the 34 cell types and derive the Markov kernel $K$ directly from the lineage graph. At each timestep, the observed population is aggregated to obtain empirical densities $\hat{\rho}_t$ over the 34 cell types. The task is to predict the population-level cell-type distribution at held-out timepoints from the observed snapshots.

### 6.3.2. MODEL

We closely follow the model design of the synthetic experiments, with two modifications to better adapt to the considered single-cell data. First, as developmental dynamics are not stationary, we follow (Terpin et al., 2024) and replace the fixed potential $V$ with a time-varying potential $V(\cdot, t)$ parametrized by an additional time input to the network. The same change is applied to the noise parameter, yielding a time-varying free-energy functional $\mathcal{F}_t(\rho) = \mathcal{V}_t(\rho) + \beta_t \mathcal{H}(\rho)$. The training objective and the geodesic-based loss are otherwise unchanged. Second, to better address boundary-approaching geodesic trajectories, we use a regularized variant of the solver compared to the naive one used in synthetic experiments: the full description is given in Appendix E.5.

### 6.3.3. RESULTS

We compare our method against the Direct-FP baseline introduced in Section 6.1.2, evaluated in the same time-varying regime for fairness. Our method achieves a Hellinger distance of 0.06, compared to 0.36 for the baseline.

We view this result as a promising indication that our framework translates beyond synthetic graphs to a non-trivial real-world setting in a prominent optimal transport application domain. Existing trajectory inference approaches typically make the problem continuous by projecting gene counts into a PCA space, while operating directly in discrete cell-type space has remained largely out of reach;

our pipeline provides an initial yet theoretically grounded computational route. We emphasize that this experiment is intentionally restricted to coarse cell-type features and is not yet applicable directly at the gene level. The associated scaling considerations are discussed in Section 7.

*Table 2.* Hellinger aggregated across all synthetic graph classes, per $\beta$.

| $\beta$ | Ours | OpenFIM |
|---|---|---|
| $\beta = 0.01$ | $0.066 \pm 0.031$ | $0.142 \pm 0.048$ |
| $\beta = 0.10$ | $0.059 \pm 0.029$ | $0.150 \pm 0.061$ |
| $\beta = 0.20$ | $0.069 \pm 0.031$ | $0.159 \pm 0.097$ |

### 6.4. Ablations

We complement the baseline comparison with a set of targeted ablations, aimed at assessing how the proposed method behaves as features such as graph size and sample availability are varied, as well as qualitatively studying the dependence on the chosen time discretization grid as well as the underlying free-energy functional. We report all details in Appendix E.4, and point the reader to Figure 3 and Figure 2.

## 7. Limitations

We emphasize an important difference between our learning assumptions and those commonly made in CTMC-based discrete diffusion generative models. In the generative setting (Austin et al., 2023; Campbell et al., 2022; Lou et al., 2024; Davis et al., 2024), one specifies a *known* forward noising process (typically a time-rescaled discrete heat equation on a graph, which for sequence generation is usually the Hamming graph with $n^d$ states, where $n$ is the alphabet size and $d$ the sequence length). Access to this ground-truth noising CTMC makes it possible to compute analytically the *conditional* vector fields (or probability ratios) required to reverse the process, enabling conditional score matching and avoiding estimation of intractable *joint* ratios in very high-dimensional spaces (Ho et al., 2020; Song et al., 2020; Gat et al., 2024).

By contrast, our framework makes the weaker assumption of observing only *temporal snapshots* along an unknown trajectory, without access to the ground-truth noising process itself. This difference is not a fundamental limitation of the geometric setup, but an assumption mismatch that induces a computational tradeoff: without the GT noising process, we cannot leverage conditional score-matching reductions, and instead resort to estimating densities from snapshots (feasible at our scale) and recover *joint* score-like quantities through the $W_K$ geometry via our quadratic geodesic routine. This is precisely why we do not run large-scale

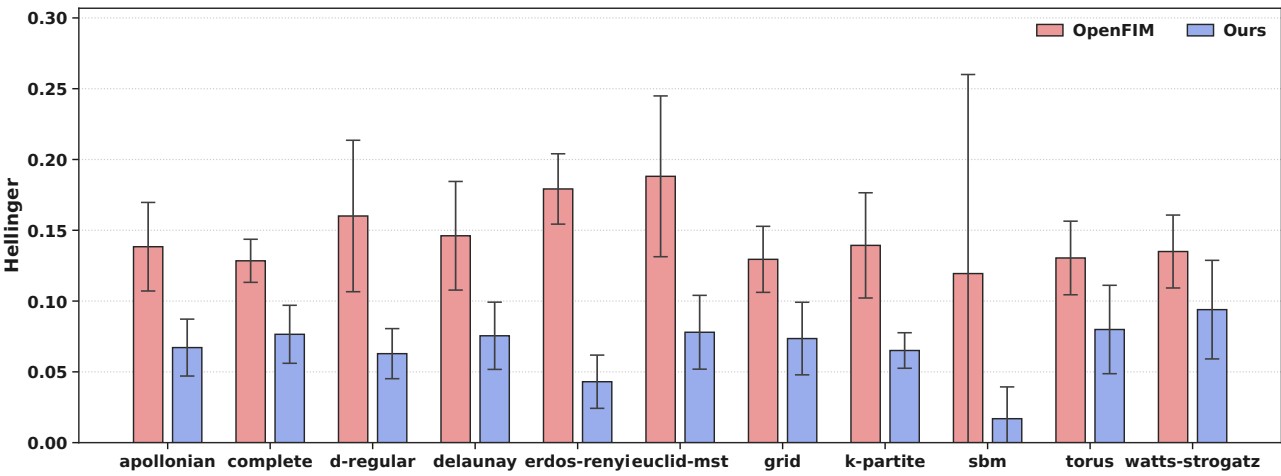

Figure 1. Hellinger distance of our model against OpenFIM, averaged across all $\beta$ levels: error bars computed over 5 samples for each configuration.

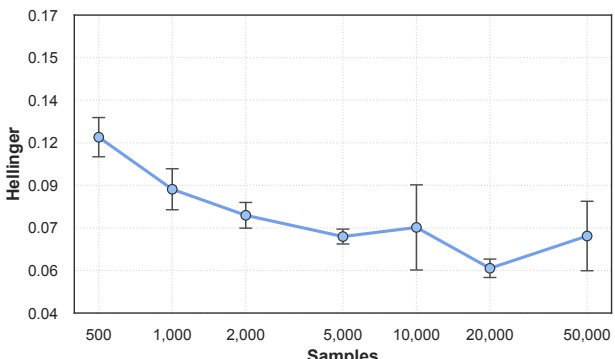

Figure 2. Hellinger distance as the number of samples increases. Averaged across all graph classes, with fixed size 10.

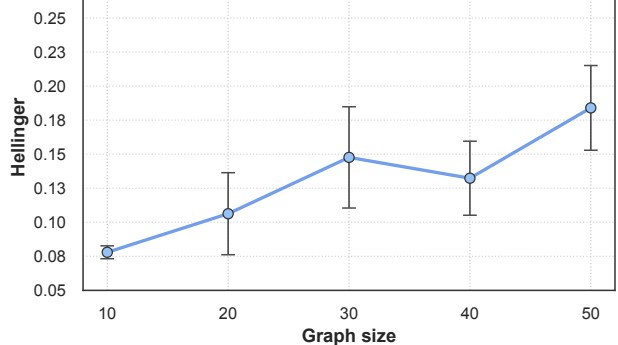

Figure 3. Hellinger distance as the graph size grows, averaged across all graph classes.

sequence-generation experiments: in language-style settings the state space is the Hamming graph with $n^d$ vertices (e.g. $n = 50257$ for GPT2 (Radford et al., 2019)), where joint density estimation from snapshots is not computationally viable.

Importantly, this does *not* break the conceptual link with discrete diffusion. When $\mathcal{F}$ is the relative entropy, the gradient in our metric satisfies $\operatorname{grad} \mathcal{H}(\rho) = \nabla \log \rho$, which coincides with the log-ratio structure that appears in time reversal of the discrete heat equation: our geodesic vectors provide a geometric route to the same object. We view integrating the computational machinery of state-of-the-art discrete diffusion models as a prominent next step. However, doing so would require a number of design choices (such as choosing conditional network outputs, tuning their expressivity, and optimizing for sparse graphs such as the Hamming graph) that are orthogonal to the foundational theory presented here. We thus leave this integration to future work, while noting that the core theoretical insights (iden-

tification of classical trajectories as gradient flows of free energies) and our practical pipeline remain fully applicable within the regimes considered in this paper.

## 8. Conclusion

In this paper, we presented a method to learn the functionals underlying discrete gradient flow dynamics. We provided a concise synthesis of the existing purely mathematical theory on the subject tailoring it to machine learning use, and developed a novel method to leverage and implement this theory to train a machine learning model. This is constituted of a numerical geodesic estimator and a quadratic-loss training routine based on imposing first-order optimality conditions. We benchmark our method on a variety of graph classes and regimes, outperforming existing baselines and ablating the role of the different components of the method such as graph structure, chosen functional and time grid.

## Acknowledgements

The authors thank Antonio Terpin for a useful discussion in the early stages of the project. DR is financed by the project "Building Energy Systems on causal reasoning (BOSS)", funded within the "Technologies and Innovations for the Climate-Neutral City" (TIKS) Programme of the Austrian Research Promotion Agency (FFG).

## Impact Statement

This paper presents work whose goal is to advance the field of Machine Learning. There are many potential societal consequences of our work, as optimal transport on discrete domains has a vast range of applications, but none of these we feel must be specifically highlighted here.

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

# A. Background

In this section, we provide some background on the various areas of mathematics and machine learning we touch in this paper, to provide the needed context on some of the tools we leverage in this work.

## A.1. Finite-State Markov Chains

Let $\mathcal{X} = \{x_1, \ldots, x_N\}$ be a finite state space and let $K \in \mathbb{R}^{N \times N}$ be a transition matrix. We say that $K$ is *row-stochastic* if

$$K_{ij} \geq 0 \ \forall i, j, \qquad \text{and} \qquad \sum_{j=1}^{N} K_{ij} = 1 \ \forall i,$$

so that each row of $K$ defines a probability distribution over next states. A vector $\pi \in \mathbb{R}^N$ is a *probability distribution* if $\pi_i \geq 0$ and $\sum_i \pi_i = 1$. It is called a *stationary distribution* for $K$ if it satisfies the invariance relation

$$\pi^\top K = \pi^\top,$$

i.e. if $\pi$ is a left eigenvector of $K$ associated with eigenvalue 1.

The matrix $K$ is said to be *irreducible* if for every pair of states $(i, j)$ there exists an integer $m \geq 1$ such that

$$(K^m)_{ij} > 0.$$

Equivalently, $K$ is irreducible if the directed graph associated with $K$ is strongly connected, where one places a directed edge $i \to j$ whenever $K_{ij} > 0$ (Golub & van Loan, 2013, Thm. 8.4.4).

The following result, known as the Perron-Frobenius Theorem, is the central result to the theory of irreducible matrices.

**Theorem A.1.** *Perron-Frobenius, (Golub & van Loan, 2013, Chapter 8) Let $A \in \mathbb{R}^{n \times n}$ be an $n \times n$ matrix with nonnegative entries, and suppose there exist $N$ such that $A^N$ has all strictly positive entries. Let $\lambda_A := \max\{|\lambda_1|, |\lambda_2|, \ldots, |\lambda_n|\}$ be the spectral radius of $A$, where $\lambda_i$ are the (complex) eigenvalues of $A$. Then the following statements hold:*

1. *$\lambda_A$ is an eigenvalue of $A$;*

2. *$\lambda_A$ is a simple eigenvalue for $A$;*

3. *there exists an eigenvector $v$ for $\lambda_A$ with strictly positive entries;*

4. *$v$ and scalar multiples are the only eigenvectors of $A$ with all entries strictly positive.*

By the Perron-Frobenius theorem, any irreducible stochastic matrix admits a unique stationary distribution $\pi$ with strictly positive entries, i.e. $\pi_i > 0$ for all $i$.

Finally, we say that $K$ is *reversible* with respect to a probability distribution $\pi$ if the *detailed balance equations* hold:

$$\pi_i K_{ij} = \pi_j K_{ji} \qquad \forall i, j.$$

In this case, $\pi$ is necessarily stationary. Indeed, summing the detailed balance relation over $i$ yields

$$\sum_{i=1}^{N} \pi_i K_{ij} = \sum_{i=1}^{N} \pi_j K_{ji} = \pi_j \sum_{i=1}^{N} K_{ji} = \pi_j,$$

which is exactly $\pi^\top K = \pi^\top$.

## A.2. Optimal transport and Gradient Flows in $\mathbb{R}^d$

We briefly recall some of the optimal transport concepts that motivate our approach; we refer to Ambrosio et al. (2008); Santambrogio (2015) for comprehensive treatments.

**Definition A.2.** Let $\mu, \nu$ be probability measures on $\mathbb{R}^d$ with finite second moments, i.e. $\mu, \nu \in \mathcal{P}_2(\mathbb{R}^d)$. The *quadratic optimal transport cost* between $\mu$ and $\nu$ is defined by

$$W_2(\mu, \nu)^2 \;:=\; \inf_{\gamma \in \Pi(\mu, \nu)} \int_{\mathbb{R}^d \times \mathbb{R}^d} \|x - y\|_2^2 \, \mathrm{d}\gamma(x, y), \tag{22}$$

where $\Pi(\mu, \nu)$ denotes the set of couplings (transport plans) with marginals $\mu$ and $\nu$.

The value $W_2$ is the *2-Wasserstein distance*, which metrizes weak convergence of measures together with convergence of second moments, and equips $\mathcal{P}_2(\mathbb{R}^d)$ with a rich geometric structure (Ambrosio et al., 2008; Santambrogio, 2015).

A key insight is that the $W_2$ distance admits a dynamical characterization in terms of minimal kinetic energy. Given $\mu_0, \mu_1 \in \mathcal{P}_2(\mathbb{R}^d)$, the Benamou–Brenier formula (Benamou & Brenier, 2000; Santambrogio, 2015) holds:

$$W_2(\mu_0, \mu_1)^2 \;=\; \inf_{\mu_t, v_t} \left\{ \int_0^1 \int_{\mathbb{R}^d} \|v_t(x)\|_2^2 \, \mathrm{d}\mu_t(x) \, \mathrm{d}t \right\}, \tag{23}$$

where the infimum is taken over time-dependent measures $(\mu_t)_{t \in [0,1]} \subset \mathcal{P}_2(\mathbb{R}^d)$ and velocity fields $(v_t)_{t \in [0,1]}$ satisfying the continuity equation (in the sense of distributions):

$$\partial_t \mu_t + \nabla \cdot (\mu_t v_t) = 0, \qquad \mu_{t=0} = \mu_0, \;\; \mu_{t=1} = \mu_1. \tag{24}$$

This formulation reveals that the Wasserstein distance can be interpreted as the minimal transport energy required to deform $\mu_0$ into $\mu_1$ along admissible mass-preserving flows.

A second fundamental insight is that several diffusion-type PDEs can be seen as *gradient flows of functionals on probability space* with respect to the $W_2$ geometry (Ambrosio et al., 2008). This is typically formalized using the Jordan–Kinderlehrer–Otto (JKO) minimizing-movement scheme (Jordan et al., 1998; Santambrogio, 2015): given an initial condition $\mu_0 \in \mathcal{P}_2(\mathbb{R}^d)$ and a time step $\tau > 0$, define the discrete-time sequence

$$\mu_{t+1} \;:=\; \arg \min_{\mu \in \mathcal{P}_2(\mathbb{R}^d)} \left\{ \mathcal{F}(\mu) + \frac{1}{2\tau} W_2(\mu, \mu_t)^2 \right\}. \tag{25}$$

Under suitable conditions on $\mathcal{F}$, the piecewise-constant interpolation of $\{\mu_t\}_t$ converges as $\tau \to 0$ to a continuous curve $(\mu_t)_{t \geq 0}$ solving the Wasserstein gradient flow associated with $\mathcal{F}$. In particular, for the *free energy* functional:

$$\mathcal{F}(\mu) = \int_{\mathbb{R}^d} V(x) \, \mathrm{d}\mu(x) + \beta \int_{\mathbb{R}^d} \rho(x) \log \rho(x) \, \mathrm{d}x \qquad (\mu = \rho \, \mathrm{d}x, \; \beta > 0),$$

the corresponding gradient flow recovers the Fokker–Planck equation. This variational viewpoint is central to modern approaches that reinterpret diffusions as steepest descent in probability space, and will serve as the blueprint for our discrete counterpart based on the metric $W_K$.

### A.3. Discrete diffusion models via continuous-time Markov chains (CTMCs)

We recall the standard setup of CTMC-based discrete diffusion models, following Campbell et al. (2022); Lou et al. (2024); Austin et al. (2023). The goal of generative modelling is to sample from an unknown data distribution $p_{\mathrm{data}}$ on a finite space $\mathcal{X}$ (in the common application of sequence generation for language/molecule domains, sequences of length $d$ over an alphabet of size $n$). Discrete diffusion constructs a *forward noising process* that pushes $p_{\mathrm{data}}$ toward a simple base distribution $\pi$ then learns to reverse this process to generate new samples.

A time-inhomogeneous CTMC is specified by a (generally time-dependent) rate matrix $Q_t$ on $\mathcal{X}$. If $p_t$ denotes the law of the process at time $t$, then it evolves by the forward Kolmogorov equation:

$$\frac{\mathrm{d}}{\mathrm{d}t} p_t \;=\; p_t Q_t, \qquad p_0 = p_{\mathrm{data}}. \tag{26}$$

A common choice for this process is the heat equation on a graph, where $Q_t = \beta_t Q$ is a scalar multiple of a fixed state transition matrix $Q$; see Campbell et al. (2022); Lou et al. (2024). The scalar parameter $\beta_t$ is typically fixed to have an

exponential schedule, and ensure fast mixing of the states that prevents the learned model to exhibit mode collapse and guarantees other stability properties (Song & Ermon, 2019).

The key fact enabling training discrete diffusion models is that the time reversal of a CTMC is again a CTMC, with reverse rates determined by $p_t$ and given by:

$$Q_t^{\text{rev}}(x, y) \; = \; Q_t(y, x) \frac{p_t(y)}{p_t(x)}. \tag{27}$$

Here a time reversal of a process with temporal law $p_t$ in the time horizon $[0, T]$ means a process with a law $q_t$ such that $p_t = q_{T-t}$. Thus learning the reverse process amounts to estimating *probability ratios* $p_t(y)/p_t(x)$ (or their logarithms) along edges where $Q_t(y, x) > 0$. In the discrete diffusion literature these log-ratios (or closely related parametrizations) play the role of a *discrete score* driving the reverse drift (Campbell et al., 2022; Austin et al., 2023; Lou et al., 2024).

Directly estimating the *joint* ratios in (27) is statistically and computationally prohibitive when $\mathcal{X}$ is huge (e.g. $|\mathcal{X}| = n^d$ for sequences). Discrete diffusion avoids this difficulty by leveraging that the *forward* corruption process is known and can be sampled: one draws $x_0 \sim p_{\text{data}}$, then simulates $x_t \sim p(\cdot \mid x_0, t)$ using the CTMC transition mechanism.

Since the forward process is known, learning is instead formulated in terms of *conditional* objectives that avoid estimating the joint ratios $p_t(y)/p_t(x)$ explicitly. Concretely, one samples $x_0 \sim p_{\text{data}}$ and then generates $x_t \sim p(\cdot \mid x_0, t)$ by simulating the forward CTMC. Using Bayes' rule, the reverse rates in (27) can be expressed in terms of conditional distributions of the form $p(x_0 \mid x_t, t)$ or, equivalently, local conditional ratios on the edges of the underlying graph.

In practice, models are trained to approximate these conditional objects from $(x_t, t)$ using denoising-style losses. A common choice, following Campbell et al. (2022), is a conditional cross-entropy objective

$$\mathcal{L}_{\text{CE}} \; = \; \mathbb{E}_{t, x_0, x_t} \big[ - \log p_\theta(x_0 \mid x_t, t) \big], \tag{28}$$

where $x_0 \sim p_{\text{data}}$ and $x_t \sim p(\cdot \mid x_0, t)$. In Hamming-graph settings, this loss is typically factorized token-wise, yielding a tractable objective even when $|\mathcal{X}| = n^d$.

An alternative but closely related formulation, proposed by Lou et al. (2024), directly targets the local reverse-time drift through a *score-entropy* loss. In this case, the model predicts a discrete score $s_\theta(x_t, t)$ on graph edges, and training minimizes an objective of the form

$$\mathcal{L}_{\text{SE}} \; = \; \mathbb{E}_{t, x_t} \Big[ \sum_{y: Q_t(x_t, y) > 0} Q_t(x_t, y) \big( s_\theta(x_t, y, t) - \log \tfrac{p_t(y)}{p_t(x_t)} \big)^2 \Big], \tag{29}$$

where the target log-ratios are computable analytically from the known forward noising process. Both objectives yield consistent estimators of the reverse-time dynamics and allow sampling by numerically simulating the learned reverse CTMC.

## B. Geometry of Discrete Probability Spaces

In this section, we give extensive details on the mathematical setup we leverage throughout the paper to describe the geometry of discrete spaces. This framework was introduced in the mathematics literature around 2011 for continuous-time Markov chains (Maas, 2011), reaction-diffusion systems (Mielke, 2011) and discretizations of Fokker-Planck equations (Chow et al., 2012).

We denote by

$$\mathcal{P}(\mathcal{X}) := \left\{ \rho : \mathcal{X} \to \mathbb{R} \;\; \rho(x) \geq 0 \;\; \forall x \in \mathcal{X}, \;\; \sum_{x \in \mathcal{X}} \pi(x) \rho(x) = 1 \right\} \tag{30}$$

the set of all *probability densities* on $\mathcal{X}$, and by $\mathcal{P}_*(\mathcal{X}) \subseteq \mathcal{P}(\mathcal{X})$ the subset of densities that are strictly positive. We choose to work with densities rather than probabilities to be more faithful to the original setting of Maas (2011): in practice, this is a simple rescaling with no difficulties, as $\pi$ is often simple and computing its value on a sequence $x$ is trivial. Throughout the

paper, we use $\rho$ to denote densities and $p$ to denote probabilities. We define the *relative entropy* of $\rho \in \mathcal{P}(\mathcal{X})$ with respect to $\pi$ as $\mathcal{H} : \mathcal{P}(\mathcal{X}) \to \mathbb{R}$ as:

$$\mathcal{H}(\rho) := \sum_{x \in \mathcal{X}} \rho(x) \log \rho(x) \pi(x).$$

It turns out that in order to define a suitable metric on the space $\mathcal{P}(\mathcal{X})$, one needs first to choose a particular way of "mixing" probabilities of individual points along edges of the graph via a function $m : \mathbb{R}_+ \times \mathbb{R}_+ \to \mathbb{R}_+$. The choice that ends up working best for our purposes is the *logarithmic mean*:

$$m(s,t) := \int_0^1 s^\alpha t^{1-\alpha} \mathrm{d}\alpha = \begin{cases} \frac{s-t}{\log(s)-\log(t)} & s \neq t \\ s & s = t \end{cases} \tag{31}$$

The choice of $m$ directly influences the functional for which we are able to identify the heat equation as the gradient flow of, and the logarithmic mean is linked to the relative entropy functional $\mathcal{H}$. Other choices that lead to similar theories with different functionals are the geometric mean $\sqrt{st}$ and, more generally, $s^\alpha t^\alpha$ for $\alpha > 0$. We refer the reader to Maas (2011); Erbar & Maas (2012b); Erbar et al. (2018) for a detailed breakdown of the conditions needed on $m$ and the resulting theories.

We use $m$ to define, for $\rho \in \mathcal{P}(\mathcal{X})$ and $x, y \in \mathcal{X}$, the density $\rho$ mixed along the edge $(x, y)$:

$$m_\rho(x,y) := m(\rho(x), \rho(y)). \tag{32}$$

We are ready to define the discrete analogous of the Wasserstein distance in the discrete space. This metric is constructed via a discrete analogous of the Benamou-Brenier (Benamou & Brenier, 2000) formula:

**Definition B.1** (Discrete metric). For $\rho_0, \rho_1 \in \mathcal{P}(\mathcal{X})$ we define:

$$W_K(\rho_0, \rho_1)^2 := \inf_{\rho, \psi} \left\{ \frac{1}{2} \int_0^1 \sum_{x,y \in \mathcal{X}} (\psi_t(x) - \psi_t(y))^2 K(x,y) m_{\rho_t}(x,y) \pi(x) \mathrm{d}t \right\}$$

where the infimum is taken over all piecewise $C^1$ curves $\rho : [0,1] \to \mathcal{P}(\mathcal{X})$ and measurable $\psi : [0,1] \to \mathbb{R}^{\mathcal{X}}$ satisfying a.e. time $t$ the *discrete continuity equation*:

$$\begin{cases} \dot{\rho}_t(x) + \sum_{y \in \mathcal{X}} (\psi_t(y) - \psi_t(x)) K(x,y) m_{\rho_t}(x,y) = 0 \\ \rho(0) = \rho_0, \ \ \rho(1) = \rho_1. \end{cases} \tag{33}$$

We highlight similarities and differences between the metric $W_K$ and the classical Wasserstein metric in the continuous case: they both can be interpreted as a "transport cost" between $\rho_0$ and $\rho_1$. However, one substantial difference is highlighted by the fact that the discrete distance of transporting a unit mass from $x$ to $y$ depends on the mass already present at $x$ and $y$. As a result, some *non-local* behavior is introduced compared to the continuous case. The discrete metric is essentially constructed as a discrete analogue of the Benamou-Brenier formula in the continuous case.

We cite the following:

**Theorem B.2** ((Maas, 2011), Thm 1.1). $W_K$ *defines a metric on* $\mathcal{P}(\mathcal{X})$.

We have now endowed $(\mathcal{P}(\mathcal{X}), W_K)$ with a metric space structure. In order to describe gradient flows on such a space, we will need to go further and endow its subspace $\mathcal{P}_*(\mathcal{X})$ with a Riemannian metric structure, similar to what has been done (at least in a formal sense) by (Otto, 2001) for the continuous case. To do so, we start with some definitions of this "discrete geometry" we will leverage throughout the paper.

**Definition B.3** (Discrete gradient and divergence). Let $\psi : \mathcal{X} \to \mathbb{R}$ (we will alternatively just write $\psi \in \mathbb{R}^{\mathcal{X}}$). We define the *discrete gradient of* $\psi$ as $\nabla \psi \in \mathbb{R}^{\mathcal{X} \times \mathcal{X}}$ given by:

$$\nabla \psi(x,y) := \psi(x) - \psi(y). \tag{34}$$

Notice that, as a special case, using this definition to write the score $\nabla \log \rho(x)$ we can draw a direct connection between the usual continuous definition of the score and the log of the ratios usually learned in discrete generative modelling. Similarly, for $\Psi \in \mathbb{R}^{\mathcal{X} \times \mathcal{X}}$ we define the *discrete divergence* $\nabla \cdot \Psi \in \mathbb{R}^{\mathcal{X}}$ as:

$$(\nabla \cdot \Psi)(x) := \frac{1}{2} \sum_{y \in \mathcal{X}} K(x,y)(\Psi(y,x) - \Psi(x,y)). \tag{35}$$

Let's also define the following scalar products for $\phi, \psi \in \mathbb{R}^{\mathcal{X}}$ and $\Phi, \Psi \in \mathbb{R}^{\mathcal{X} \times \mathcal{X}}$:

$$\langle \phi, \psi \rangle_\pi := \sum_{x \in \mathcal{X}} \phi(x)\psi(x)\pi(x), \tag{36}$$

$$\langle \Phi, \Psi \rangle_\pi = \frac{1}{2} \sum_{x,y \in \mathcal{X}} \Phi(x,y)\Psi(x,y)K(x,y)\pi(x). \tag{37}$$

Notice that the last one is a proper scalar product because of the reversibility hypothesis on the chain. Finally, for $\rho \in \mathcal{P}(\mathcal{X})$ define:

$$\langle \Phi, \Psi \rangle_\rho := \frac{1}{2} \sum_{x,y \in \mathcal{X}} \Phi(x,y)\Psi(x,y)K(x,y)m_\rho(x,y)\pi(x), \tag{38}$$

together with the associated norm:

$$\|\Phi\|_\rho^2 := \langle \Phi, \Phi \rangle_\rho. \tag{39}$$

One can easily check that the following *"integration by parts"* formula holds:

**Proposition B.4** (Discrete integration by parts)**.** *Let $\psi \in \mathbb{R}^{\mathcal{X}}$ and $\Psi \in \mathbb{R}^{\mathcal{X} \times \mathcal{X}}$. Then:*

$$\langle \nabla \psi, \Psi \rangle_\pi = - \langle \psi, \nabla \cdot \Psi \rangle_\pi. \tag{40}$$

*Proof.* Straightforward from the definitions. $\qquad\square$

Furthermore, let's denote the element-wise product of two elements $M, N \in \mathbb{R}^{\mathcal{X} \times \mathcal{X}}$ as:

$$(M \odot N)(x,y) := M(x,y)N(x,y). \tag{41}$$

*Remark* B.5. We can reformulate the discrete continuity equation (5) as:

$$\dot\rho_t + \nabla \cdot (m_{\rho_t}(x,y) \odot \nabla \psi_t) = 0, \tag{42}$$

which highlights the connection between this discrete equation and its classical continuous counterpart.

In order to construct a Riemannian metric on $(\mathcal{P}_*(\mathcal{X}), W_K)$, we will use as the tangent space to a point $\rho \in \mathcal{P}_*(\mathcal{X})$ the collection of discrete gradients:

$$T_\rho := \left\{ \nabla \psi \in \mathbb{R}^{\mathcal{X} \times \mathcal{X}} : \psi \in \mathbb{R}^{\mathcal{X}} \right\}. \tag{43}$$

We can do this in light of the following result on the structure of discrete tangent vectors along curves, which we state informally without the mathematical machinery built by (Maas, 2011). The reader can consult the original source for a formal derivation.

**Proposition B.6** ((Maas, 2011), Prop. 3.26, informal)**.** *Let $\rho_t$ be a smooth curve in $\mathcal{P}_*(\mathcal{X})$ defined for $t \in [0,1]$, and take a fixed $t$. Then there exists a unique discrete gradient $\nabla \psi_t$ such that (5) holds.*

Essentially, this results justifies the intuitive notion that, if we define smooth curves in $\mathcal{P}_*(\mathcal{X})$ via the continuity equation (5), one can identify velocities (and by extension, any tangent vector) of such curves via the discrete gradients $\nabla \psi$ of real-valued functions on $\mathcal{X}$.

In light of this, we can endow $T_\rho$ with the inner product $\langle \cdot, \cdot \rangle_\rho$ given by:

$$\langle \nabla \phi, \nabla \psi \rangle_\rho = \frac{1}{2} \sum_{x,y \in \mathcal{X}} (\phi(x) - \phi(y))(\psi(x) - \psi(y)) K(x,y) m_\rho(x,y) \pi(x). \tag{44}$$

We can finally state the following, which concludes the construction of the Riemannian structure on discrete Markov chains we will leverage.

**Theorem B.7** ((Maas, 2011), Thm. 3.29). *The space $(\mathcal{P}_*(\mathcal{X}), W_K)$ is a Riemannian manifold. The Riemannian product at a point $\rho$ is given by* (8), *and it induces the distance $W_K$ as a Riemannian distance on $\mathcal{P}_*(\mathcal{X})$.*

We also cite the following result from Maas (2011); Erbar & Maas (2012a), which provides a discrete version of the geodesic equations for the metric $W_K$, at least locally.

**Proposition B.8** ((Maas, 2011), Thm. 3.31, (Erbar & Maas, 2012a) Prop. 3.4). *Let $\rho \in \mathcal{P}_*(\mathcal{X})$ and let $\psi : \mathcal{X} \to \mathbb{R}$. On a small enough time interval around $0$, the unique constant speed geodesic with $\rho_0 = \rho$ and initial velocity $\nabla \psi_0 = \nabla \psi$ satisfies:*

$$\begin{cases} \partial_t \rho_t(x) + \sum_{y \in \mathcal{X}} (\psi_t(y) - \psi_t(x)) K(x,y) m_{\rho_t}(x,y) = 0 \\ \partial_t \psi_t(x) + \frac{1}{2} \sum_{y \in \mathcal{X}} (\psi_t(y) - \psi_t(x))^2 K(x,y) \partial_1 m_{\rho_t}(x,y) = 0. \end{cases} \tag{45}$$

The main result we will leverage will concern the gradient flow of a heat-like equation in $\mathcal{P}_*(\mathcal{X})$.

Let $\rho : (0, \infty) \to (\mathcal{P}_*(\mathcal{X}), W_K)$ be a smooth curve, and let $D_t \rho$ denote the tangent vector to this curve at the point $\rho_t$. Furthermore, let $F : \mathcal{P}_*(\mathcal{X}) \to \mathbb{R}$ be a smooth functional, and let $\operatorname{grad} F$ denote its gradient.

The main results concerning gradient flow trajectories in this space is the following:

**Theorem B.9** ((Maas, 2011), Thm. 4.7). *Let $\rho_0 \in \mathcal{P}(\mathcal{X})$, $t \geq 0$, and define the heat equation as*

$$\rho_t := e^{t(K-I)} \rho_0. \tag{46}$$

*Let $\mathcal{H} : \mathcal{P}_*(\mathcal{X}) \to \mathbb{R}$ be the relative entropy functional* (9). *Then the gradient flow equation:*

$$D_t \rho = -\operatorname{grad} \mathcal{H}(\rho_t) \tag{47}$$

*holds for all $t > 0$.*

Thanks to this structure, we are in a position to study functionals on the space $\mathcal{P}_*(\mathcal{X})$ and ask what the gradient flow of a certain functional is. We now turn to do that and show how to construct a loss function with these tools.

## C. Proofs

### C.1. Divergence of heat equation velocity in the $W_2$ space

**Lemma C.1.** *Let $\mathcal{X} := \{a, b\}$ be a 2-point space, and let $K$ be a Markov kernel on $\mathcal{X}$ with $K(a,b) = p$, $K(b,a) = q$, $0 < p, q < 1$, which has invariant probability $\pi = \left[ \frac{q}{p+q}, \frac{p}{p+q} \right]$. Let $p \neq \pi$ be any non-invariant probability on $\mathcal{X}$, and let $\rho$ be the unique density associated to the probability $p$, which satisfies $p = \rho \odot \pi$. Let $\rho_t$ denote the solution of the discrete heat equation* (2) *with $\rho_0 = \rho$, and $p_t = \rho_t \odot \pi$ the associated probability curve. Then the metric derivative of the $W_2$ distance along $p_t$ diverges, that is:*

$$|\dot{p}_t| := \limsup_{s \to t} \frac{W_2(p_t, p_s)}{|t - s|} = +\infty.$$

*Proof.* Let's call $\mathcal{Q}$ the space of probabilities on $\mathcal{X}$. Each measure in this space is of the kind $p_\beta = \frac{1}{2}((1-\beta)\delta_a + (1+\beta)\delta_b)$ for a parameter $\beta \in [-1, 1]$ and $\delta_x$ a Dirac delta measure. The corresponding density $\rho_\beta$ w.r.t. $\pi$ is given by:

$$\rho_\beta(a) = \frac{p+q}{q}\frac{1-\beta}{2} \quad, \quad \rho_\beta(b) = \frac{p+q}{p}\frac{1+\beta}{2}. \tag{48}$$

Now, let $H(t) = e^{t(K-I)}$ be the heat semigroup, and let $\rho_{\beta_t} = H(t)\rho_\beta = u(t)$ be the associated heat equation starting from a certain density $\rho_\beta$ with $\beta \in [-1, 1] \setminus \{\frac{p-q}{p+q}\}$. One easily sees that the parameter $\beta_t$ depends on $\beta$ and $t$ by:

$$\beta_t := \frac{p-q}{p+q}(1 - e^{-(p+q)t}) + \beta e^{-(p+q)t}. \tag{49}$$

Now, a direct calculation shows that $W_2(\rho_\alpha, \rho_\beta) = \sqrt{2|\alpha - \beta|}$ because of the classical discrete definition of $W_2$ (Santambrogio, 2015). Then we see that:

$$\begin{aligned}
|\dot{u}(t)| :&= \limsup_{s \to t} \frac{W_2(u(t), u(s))}{|t-s|} = \sqrt{2}\limsup_{s \to t} \frac{\sqrt{|\beta_t - \beta_s|}}{|t-s|} = \\
&= \sqrt{2\left|\beta - \frac{p-q}{p+q}\right|}\limsup_{s \to t} \frac{\sqrt{|e^{-(p+q)t} - e^{-(p+q)s}|}}{|t-s|} = +\infty.
\end{aligned} \tag{50}$$

$\square$

### C.2. Gradient Flow Characterization of Useful Functionals

**Lemma C.2** (Gradient of potential energy functionals in the $W_K$ metric)**.** *Define the potential energy functional as* $\mathcal{V} : \mathcal{P}_*(\mathcal{X}) \to \mathbb{R}$ :

$$\mathcal{V}(\rho) := \sum_{x \in \mathcal{X}} V(x)\rho(x)\pi(x), \tag{51}$$

*where $V$ is a function from $\mathcal{X}$ onto $\mathbb{R}$. Then $\mathcal{V}$ is differentiable and its gradient at a point $\rho \in \mathcal{P}_*(\mathcal{X})$ is given by:*

$$\operatorname{grad} \mathcal{V}(\rho) = \nabla V. \tag{52}$$

*Proof.* Differentiability of $\mathcal{V}$ is immediate. Now, fix a curve $\rho_t$ such that $D_t \rho = \nabla \psi_t$. Then:

$$\begin{aligned}
\frac{\mathrm{d}}{\mathrm{d}t}\mathcal{V}(\rho_t) &= \sum_{x \in \mathcal{X}} V(x)\dot{\rho}_t(x)\pi(x) = \\
&= -\langle V, \nabla \cdot (m_{\rho_t}(x,y) \odot \nabla \psi_t)\rangle_\pi = \\
&= \langle \nabla V, m_{\rho_t}(x,y) \odot \nabla \psi_t\rangle_\pi = \\
&= \langle \nabla V, \nabla \psi_t\rangle_{\rho_t},
\end{aligned} \tag{53}$$

which proves the result. $\square$

**Proposition C.3** (Gradient flow of $W_K^2$)**.** *Let $\rho_t \in \mathcal{P}_*(\mathcal{X})$, let $U$ be a neighbourhood of $\rho_t$ where $\mathcal{G}$ is smooth. For a point $\rho \in U$, let $v$ be the starting velocity of the unique geodesic starting from $\rho$ and passing through $\rho_t$. The gradient flow of $\mathcal{G}$ is given by:*

$$\operatorname{grad} \mathcal{G}(\rho) = -2v. \tag{54}$$

*Proof.* Recall that we want to show that, choosing a vector $\nabla\psi \in T_{\rho_t}\mathcal{P}_*(\mathcal{X})$ and a curve $\eta(t)$ with $\eta(0) = \rho_t$ and $\dot{\eta}(0) = \nabla\psi$, then:

$$\left.\frac{d}{dt}W_K(\rho, \eta(t))^2\right|_{t=0} = \left\langle\nabla\psi, -2\exp_\rho^{-1}(\rho_t)\right\rangle_{\rho_t}. \tag{55}$$

Let's now consider such a curve $\eta$ and a smooth map $H(s, t)$ with values in $\mathcal{P}_*(\mathcal{X})$ such that $\gamma_t(\cdot) := H(\cdot, t)$ is a constant speed geodesic between $\rho$ and $\eta(t)$.

Let's define the *kinetic energy* of a smooth curve $\gamma : [0, 1] \to \mathcal{P}_*(\mathcal{X})$ as:

$$E(\gamma) := \frac{1}{2}\int_0^1 \|\dot{\gamma}(s)\|_{\gamma(s)}^2 \, ds. \tag{56}$$

Recall that the norm here is defined by (8). The Riemannian distance $W_K(\rho, \rho_t)$ can be defined as the length of a geodesic:

$$W_K(\rho, \rho_t) = \int_0^1 \|\dot{\gamma}(s)\|_{\gamma(s)} \, ds. \tag{57}$$

Now, thanks to $\gamma_t$ being constant speed it follows:

$$\left.\frac{d}{dt}W_K(\rho, \eta(t))^2\right|_{t=0} = 2\left.\frac{d}{dt}E(\gamma_t)\right|_{t=0}. \tag{58}$$

Now, using the formula for the First variation of Energy (cfr (Petersen, 2006), Lemma 5.4.2) we get:

$$\left.\frac{d}{dt}E(\gamma_t)\right|_{t=0} = \left\langle\nabla\psi, \dot{\gamma}_0(0)\right\rangle_{\rho_t}, \tag{59}$$

but $\dot{\gamma}_0(0)$ is exactly the quantity we were looking for, i.e. the initial velocity of a constant speed geodesic joining $\rho$ and $\rho_t$, which concludes the proof. $\qquad\square$

**Lemma C.4** (Characterization of the gradient of the Entropy in the $W_K$ metric)**.** *Let $\mathcal{H}$ be the relative entropy functional, and $\rho \in \mathcal{P}_*(\mathcal{X})$. Then:*

$$\operatorname{grad}\mathcal{H}(\rho) = \nabla\log\rho. \tag{60}$$

*Proof.* Recall that the general definition of the gradient of a smooth function $f : M \to \mathbb{R}$ at a point $x \in M$ of a Riemannian manifold is a vector $\operatorname{grad}f(x) \in T_xM$ such that, for every curve $\gamma : (-1, 1) \to M$ with $\gamma(0) = x$ and $\gamma'(0) = v \in T_xM$:

$$\left.\frac{\mathrm{d}}{\mathrm{d}t}f(\gamma(t))\right|_{t=0} = \left\langle\operatorname{grad}f(x), v\right\rangle_x \tag{61}$$

according to the scalar product defined on $T_xM$. Fix a curve $\rho_t$ such that $D_t\rho = \nabla\psi_t$, that is:

$$\dot{\rho}_t + \nabla \cdot (m_{\rho_t}(x, y) \odot \nabla\psi_t) = 0. \tag{62}$$

Substituting the entropy in the gradient definition we get:

$$\begin{aligned}
\frac{\mathrm{d}}{\mathrm{d}t}\mathcal{H}(\rho_t) &= \sum_{x\in\mathcal{X}}(\log(\rho_t) + 1)\dot{\rho}_t\pi(x) = \\
&= -\sum_{x\in\mathcal{X}}(\log(\rho_t) + 1)\nabla \cdot (m_{\rho_t}(x, y) \odot \nabla\psi_t)\pi(x) = \\
&= -\left\langle\log(\rho_t) + 1, \nabla \cdot (m_{\rho_t}(x, y) \odot \nabla\psi_t)\right\rangle_\pi
\end{aligned} \tag{63}$$

Integrating by parts and using the definition of $\langle \cdot, \cdot \rangle_{\rho_t}$:

$$\frac{\mathrm{d}}{\mathrm{d}t}\mathcal{H}(\rho_t) = -\langle \log(\rho_t) + 1, \nabla \cdot (m_{\rho_t}(x, y) \odot \nabla \psi_t)\rangle_\pi =$$
$$= \langle \nabla[\log(\rho_t) + 1], m_{\rho_t}(x, y) \odot \nabla \psi_t)\rangle_\pi = \tag{64}$$
$$= \langle \nabla[\log(\rho_t) + 1], \nabla \psi_t)\rangle_{\rho_t},$$

and since trivially $\nabla[\log(\rho_t) + 1] = \nabla \log(\rho_t)$ this completes the proof.

$\square$

### C.3. Solutions of the Discrete JKO Scheme

**Theorem C.5.** *Consider the discrete JKO scheme* (10) *for the discrete free energy functional:*

$$\rho_{t+1} = \underset{\rho \in \mathcal{P}_*(\mathcal{X})}{\arg\min} \left\{ \mathcal{F}(\rho) + \frac{1}{2\tau} W_K(\rho, \rho_t)^2 \right\} \tag{65}$$

*with* $\mathcal{F}(\rho) := \mathcal{V}(\rho) + \beta \mathcal{H}(\rho)$ *with* $\beta > 0$. *Then for any* $\tau > 0$ *this scheme has a unique minimizer* $\hat{\rho}_{t+1}$ *in* $\mathcal{P}_*(\mathcal{X})$. *If* $\hat{\rho}_{t+1}$ *is this minimizer, let* grad *be the gradient operator at the point* $\hat{\rho}_{t+1}$ *w.r.t. the* $W_K$-*induced metric. Then:*

$$\mathrm{grad}\left( \mathcal{F}(\hat{\rho}_{t+1}) + \frac{1}{2\tau} W_K(\hat{\rho}_{t+1}, \rho_t)^2 \right) = 0. \tag{66}$$

*Proof.* Fix $\tau > 0$ and $\rho_t \in \mathcal{P}_*(\mathcal{X})$. Define the objective

$$J(\rho) := \mathcal{F}(\rho) + \frac{1}{2\tau} W_K(\rho, \rho_t)^2, \qquad \rho \in \mathcal{P}(\mathcal{X}),$$

where $\mathcal{F}(\rho) = \mathcal{V}(\rho) + \beta \mathcal{H}(\rho)$ with $\beta > 0$.

**Existence of a Minimizer** Since $\mathcal{X}$ is finite, the simplex $\mathcal{P}(\mathcal{X})$ is compact in $\mathbb{R}^\mathcal{X}$. The functional $\mathcal{V}$ is continuous on $\mathcal{P}(\mathcal{X})$ and $\mathcal{H}$ is lower semicontinuous (with the convention $0 \log 0 := 0$), hence $\mathcal{F}$ is lower semicontinuous. Moreover, $\rho \mapsto W_K(\rho, \rho_t)^2$ is continuous on $\mathcal{P}(\mathcal{X})$ since $\rho_t \in \mathcal{P}_*(\mathcal{X})$ and $W_K$ metrizes weak convergence on $\mathcal{P}_*(\mathcal{X})$ since all topologies coincide on a finite space. Therefore $J$ is lower semicontinuous on the compact set $\mathcal{P}(\mathcal{X})$, and admits at least one minimizer $\hat{\rho}_{t+1} \in \mathcal{P}(\mathcal{X})$.

**The minimizer is in the interior $\mathcal{P}_*(\mathcal{X})$** Assume by contradiction that $\hat{\rho}_{t+1} \notin \mathcal{P}_*(\mathcal{X})$. Then there exists $x_0 \in \mathcal{X}$ such that $\hat{\rho}_{t+1}(x_0) = 0$. For $\varepsilon \in (0, 1)$, consider the linear interpolation

$$\rho_\varepsilon := (1 - \varepsilon)\hat{\rho}_{t+1} + \varepsilon \rho_t.$$

Then $\rho_\varepsilon \in \mathcal{P}_*(\mathcal{X})$ for all $\varepsilon > 0$ since $\rho_t \in \mathcal{P}_*(\mathcal{X})$. Using $\rho_\varepsilon(x_0) = \varepsilon \rho_t(x_0)$ and $s \log s \to 0$ as $s \downarrow 0$, we have

$$\mathcal{H}(\rho_\varepsilon) = \mathcal{H}(\hat{\rho}_{t+1}) + \pi(x_0)\varepsilon \rho_t(x_0) \log(\varepsilon \rho_t(x_0)) + O(\varepsilon),$$

hence

$$\mathcal{H}(\rho_\varepsilon) \le \mathcal{H}(\hat{\rho}_{t+1}) + c\varepsilon \log \varepsilon \qquad \text{for } \varepsilon \text{ small enough,}$$

for some $c > 0$. Since $\mathcal{V}$ is affine in $\rho$, we also have $\mathcal{V}(\rho_\varepsilon) = \mathcal{V}(\hat{\rho}_{t+1}) + O(\varepsilon)$, hence, using $\beta > 0$,

$$\mathcal{F}(\rho_\varepsilon) \le \mathcal{F}(\hat{\rho}_{t+1}) + c'\varepsilon \log \varepsilon \qquad \text{for } \varepsilon \text{ small enough,}$$

for some $c' > 0$, and in particular $\mathcal{F}(\rho_\varepsilon) < \mathcal{F}(\hat{\rho}_{t+1})$ for $\varepsilon$ small.

On the other hand, by convexity of $\rho \mapsto W_K(\rho, \rho_t)^2$ along linear interpolations (Erbar & Maas, 2012a, Prop. 2.1.1), we obtain

$$W_K(\rho_\varepsilon, \rho_t)^2 \le (1 - \varepsilon) \, W_K(\hat{\rho}_{t+1}, \rho_t)^2 + \varepsilon \, W_K(\rho_t, \rho_t)^2 = (1 - \varepsilon) \, W_K(\hat{\rho}_{t+1}, \rho_t)^2 \le W_K(\hat{\rho}_{t+1}, \rho_t)^2.$$

Combining the two inequalities yields, for $\varepsilon$ small enough,

$$J(\rho_\varepsilon) < J(\hat{\rho}_{t+1}),$$

contradicting the minimality of $\hat{\rho}_{t+1}$. Therefore any minimizer must belong to $\mathcal{P}_*(\mathcal{X})$.

**Uniqueness.** Let $\rho_0 \ne \rho_1$ in $\mathcal{P}(\mathcal{X})$ and $\rho_\lambda = (1 - \lambda)\rho_0 + \lambda\rho_1$. The map $\rho \mapsto \mathcal{V}(\rho)$ is affine, the map $\rho \mapsto W_K(\rho, \rho_t)^2$ is convex along $\rho_\lambda$ (Erbar & Maas, 2012a, Prop. 2.1.1), and the entropy $\mathcal{H}$ is strictly convex on $\mathcal{P}(\mathcal{X})$ since $s \mapsto s \log s$ is strictly convex on $\mathbb{R}_+$. Since $\beta > 0$, it follows that $J$ is strictly convex along linear interpolations, hence admits at most one minimizer. Therefore the minimizer $\hat{\rho}_{t+1}$ is unique.

**First-order condition.** By the previous steps, the unique minimizer $\hat{\rho}_{t+1}$ belongs to $\mathcal{P}_*(\mathcal{X})$. Since $\mathcal{P}_*(\mathcal{X})$ is a smooth Riemannian manifold endowed with the $W_K$-induced metric and $J$ is smooth on $\mathcal{P}_*(\mathcal{X})$, the minimizer satisfies the first-order optimality condition

$$\operatorname{grad} J(\hat{\rho}_{t+1}) = 0,$$

which is exactly (66).  □

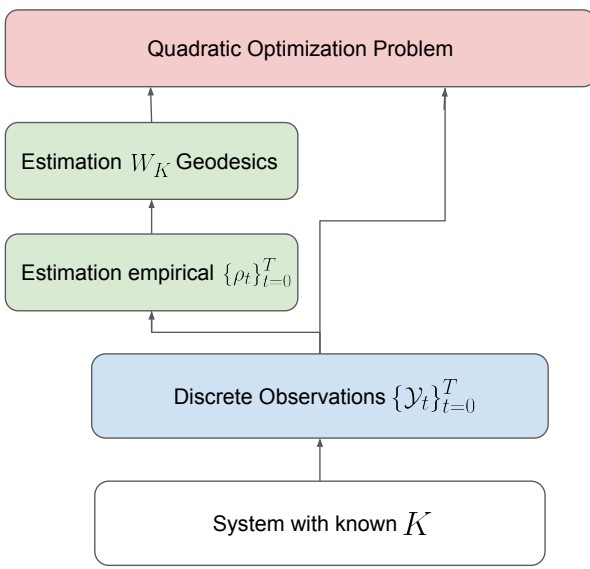

*Figure 4.* Schematic breakdown of our pipeline: we first estimate densities, use them to compute geodesics, then feed everything into the quadratic loss.

## D. Inference Details

We use the same procedure to evolve a population given a gradient of a potential $\nabla V$ and a parameter $\beta$ both in the data generation (using the ground truth values) and for inference.

Given the estimated density from the previous timestep $\hat{\rho}$, potential $V$ and parameter $\beta$, we define:

$$\psi_i \;=\; V_i + \beta \log \hat{\rho}_i,$$

and we define the off-diagonal jump rate from $i$ to $j$ as:

$$Q_{ij} \;=\; \frac{K_{ij}\, \Theta_{ij}}{\rho_i} \left[\psi_j - \psi_i\right]_+, \qquad i \neq j,$$

and the diagonal is closed as $Q_{ii} = -\sum_{j \neq i} Q_{ij}$ so that rows sum to zero, where $\Theta$ is again the log-mean matrix computed from $\hat{\rho}$. We evolve samples by freezing $Q$ on each time interval and drawing from $P = \exp(\Delta t\, Q)$ via a standard matrix exponential.

*Remark* D.1. More sophisticated approaches for solving dynamical OT problems at wider timesteps, for example see (Facca et al., 2024) for an approach using efficient preconditioners. However, we choose to keep our numerical approach as simple as possible, to show that the success of our pipeline does not rely on sophisticated numerical estimations, but the most standard approaches work well.

## E. Additional Experimental Details

### E.1. Metric

We give here a formal definition of the Hellinger distance we use throughout the paper, for completeness:

$$H(p, q) := \frac{1}{\sqrt{2}} \sqrt{\sum_{i=1}^{n} (\sqrt{p_i} - \sqrt{q_i})^2}.$$

### E.2. Optimizer

We use the Adam optimizer (Kingma & Ba, 2017) with the parameters $\beta_1 = 0.9$, $\beta_2 = 0.999$, $\epsilon = 1e^{-8}$, and constant learning rate $lr = 5e^{-4}$. We use a weight decay of $1e - 6$.

### E.3. Batch Construction and Time Sampling

The time integral appearing in the training objective is approximated via Monte Carlo sampling over time, following the standard practice in diffusion and score-based generative models. At each optimization step, training samples are paired with independently sampled time indices, yielding minibatches that contain data points evaluated at heterogeneous time steps. This batching strategy is analogous to the one used in denoising diffusion probabilistic models (DDPMs) (Ho et al., 2020) and score-based stochastic differential equation models (Song et al., 2021), as well as in their discrete counterparts (Campbell et al., 2022).

### E.4. Ablations

We complement the baseline comparison with a set of targeted ablations, aimed at assessing how the proposed method behaves as features such as graph size and sample availability are varied.

#### E.4.1. ABLATING ON GRAPH SIZE $N$

We first study the effect of increasing graph size on model performance. To this end, we train and evaluate our method on graphs of increasing size, sampling graph instances in each synthetic class with numbers of vertices $N \in \{10, 20, 30, 40, 50\}$. Performance is measured in terms of Hellinger distance and averaged across graph classes.

As shown in Figure 3, we observe a gradual degradation in performance as the graph size increases, with Hellinger distance exhibiting approximately linear growth in $N$ over the considered range. This behavior is consistent with the increasing dimensionality of the probability simplex and the corresponding difficulty of the optimization problem.

Empirically, for larger graphs we also observe occasional optimization instabilities in the quadratic objective, with around $1.5\%$ of learned distributions collapsing toward degenerate solutions concentrated on a single vertex. We conjecture that

this phenomenon is related to trajectories approaching the boundary of the probability simplex, where the underlying geometry becomes non-smooth. In the present study, we focus on stable runs to characterize typical performance, and leave a systematic treatment of these instabilities to future work. We remark that these failed runs are easy to identify from the constancy of the predicted states among elements of $\mathcal{X}$.

### E.4.2. ABLATING THE NUMBER OF SAMPLES

We next investigate the dependence of model performance on the number of observed samples per run. For this ablation, we fix the graph size to $N = 10$ and evaluate across all graph classes while varying the number of samples in $\{500, 1000, 2000, 5000, 10000\}$.

Results are reported in Figure 2. We find that performance is largely insensitive to the number of samples once a moderate threshold is reached, with Hellinger distance remaining stable across a wide range of sample sizes. This behavior is consistent with the structure of the learning objective, which operates on empirical density estimates living in an $N$-dimensional space rather than directly on individual samples.

A slight degradation in performance is observed at the smallest sample size (500), which we attribute to increased variance in the empirical estimate of the density $\hat{\rho}$. Overall, these results suggest that the proposed method does not require large sample sizes to achieve accurate recovery of the underlying dynamics.

### E.4.3. ABLATING THE TIME GRID

We qualitatively assess the sensitivity of the proposed method to the choice of temporal discretization. We consider uniform time grids, randomized irregular grids, and adaptive logarithmic schedules following Wang et al. (2025), with resolutions ranging from 20 to $10^4$ time steps in all cases.

Across all tested regimes, our method consistently outperforms the OpenFIM baseline, with an average Hellinger distance of approximately $0.11$ compared to $0.18$ for OpenFIM. Increasing the temporal resolution leads to modest but consistent improvements in performance, while differences between grid types remain small, with adaptive logarithmic schedules performing slightly better on average. Overall, these results indicate that the method is not sensitive to the precise choice of time grid within a broad and practically relevant range.

### E.4.4. ABLATING THE FREE-ENERGY FUNCTIONAL

We further perform a qualitative ablation on the structure of the free-energy functional by varying both the scale of the potential and the relative weight of the entropy term. Specifically, we consider multiple ranges for the potential amplitude together with increasing values of the entropy coefficient $\beta$.

Across the tested regimes, we observe that larger potential amplitudes tend to slightly degrade both performance and stability, while increasing the entropy weight consistently improves robustness and accuracy. In particular, the purely entropic regime corresponds to perfectly stable dynamics, with no observed optimization collapse, reflecting the well-conditioned nature of the associated heat flow.

These trends are consistent with the role of entropy as a regularizing component of the dynamics and suggest that the proposed method behaves in line with theoretical expectations. A more exhaustive quantitative exploration of these effects is left for future work.

### E.5. Regularized Geodesic Solver

The standard geodesic solver requires solving a constrained quadratic program at each consecutive pair of density snapshots $(\hat{\rho}_t, \hat{\rho}_{t+1})$. On synthetic data all populations remain strictly positive throughout training, so the solver is well-conditioned. On real biological data, however, individual cell types can be absent at certain timepoints, causing near-zero entries in the density vector and ill-conditioning in both the log-mean mobility $\Theta(\rho_i, \rho_j) = (\rho_i - \rho_j)/(\log \rho_i - \log \rho_j)$ and the resulting Laplacian. We address this with several targeted modifications.

- **Density clipping.** Before computing the log-mean mobility $\theta$, both $\hat{\rho}_t$ and $\hat{\rho}_{t+1}$ are clamped from below to $\varepsilon_\rho = 10^{-4}$, preventing zero-density nodes from causing $\log 0$ instabilities in the Laplacian.

- **Midpoint mobility.** The log-mean mobility is evaluated at the midpoint density $\frac{1}{2}(\hat{\rho}_t + \hat{\rho}_{t+1})$ rather than at $\hat{\rho}_t$ alone,

which better approximates the density along the geodesic and reduces sensitivity to vanishing endpoints.

- **Adaptive gauge pinning.** The gauge degree of freedom in the potential $\psi$ is fixed by pinning the node with the highest stationary probability $\pi_i$ to zero, rather than always using node 0. This choice yields a more numerically stable reference point on graphs with heterogeneous stationary weights.

- **Log-ratio fallback.** When the constrained Cholesky solver fails (e.g., due to a non-positive-definite system at a degenerate timepoint), we fall back to the log-ratio velocity field $v_{ij} = \log(\hat{\rho}_{t+1,j}/\hat{\rho}_{t,j}) - \log(\hat{\rho}_{t+1,i}/\hat{\rho}_{t,i})$, which remains well-defined even when densities approach the boundary of the simplex.

## F. A Guide for the Discrete Diffusion Practitioner

We summarize here how the geometric framework developed in this paper connects to (and can be used to reinterpret and extend) CTMC-based discrete diffusion models commonly used in generative modeling (Austin et al., 2023; Campbell et al., 2022; Lou et al., 2024; Davis et al., 2024).

A standard discrete diffusion model specifies a *forward* continuous-time Markov chain (CTMC) on a discrete state space $\mathcal{X}$. Concretely, one chooses a graph structure encoding which transitions are allowed and assigns jump rates so that data samples progressively evolve toward a tractable base distribution. Learning then consists of approximating the *reverse-time* process, enabling sampling from the data distribution by simulating the reverse CTMC.

A common and practically successful choice is a *time-rescaled discrete heat equation* on the state space (Campbell et al., 2022; Lou et al., 2024). In our notation, this is the density evolution:

$$\frac{\mathrm{d}}{\mathrm{d}t}\rho_t = (K - I)\rho_t, \tag{67}$$

where $K$ is a Markov kernel on $\mathcal{X}$. The kernel $K$ controls which transition probabilities are instantaneously nonzero and with what relative strength: in this sense, $K$ can be used to model between which elements of $\mathcal{X}$ the density can flow in the path from the starting point to the data distribution. Sparsity in $K$ also helps computationally (Lou et al., 2024): when applying conditional flow matching, the only non-zero elements in the conditional discrete score $\nabla \log p(x|x_0)$ are those $x$ such that $K(x, x_0) \neq 0$, which allows for example not to store any other entry of the conditional score in memory, significantly shrinking the computational burden for large $\mathcal{X}$ spaces via conditional flow matching.

In the discrete sequence generation domain, if one considers an alphabet of $n$ letters and sequences of length $d$, $|\mathcal{X}| = n^d$, and the typical choice for $K$ is the $(n, d)$ *Hamming graph*:

$$K(x, y) = \begin{cases} 0 & \text{if } d_H(x, y) > 1, \\ Q_{\text{tok}}(x_i, y_i) & \text{otherwise, with } i \text{ the mismatched position (if any)} \end{cases} \tag{68}$$

where $d_H$ indicates the Hamming distance between sequences $x$ and $y$, which counts the number of tokens (with position) the sequences differ at. $Q_{\text{tok}}$ is a tunable token-level transition matrix, which in applications is typically set to be either uniform across all states or absorbing to an external, auxiliary state (Campbell et al., 2022; Lou et al., 2024).

The central quantity in CTMC-based discrete diffusion models is the family of probability ratios appearing in the time-reversal of the forward noising process. In practice, learning the reverse dynamics can be expressed in terms of learning probability ratios (or "scores") (Austin et al., 2023; Campbell et al., 2022; Lou et al., 2024):

$$s(x) := \left[\frac{p(y)}{p(x)}\right]_{y \in \mathcal{X}}. \tag{69}$$

This score is closely related to the entropy gradient in the Maas geometry: recall the relative entropy (w.r.t. the invariant measure $\pi$ of $K$):

$$\mathcal{H}(\rho) := \sum_{x \in \mathcal{X}} \rho(x) \log \rho(x)\, \pi(x). \tag{70}$$

Then the $W_K$-gradient of $\mathcal{H}$ is, from C.4:

$$\operatorname{grad}\mathcal{H}(\rho) = \nabla \log \rho, \tag{71}$$

where $\nabla f(x, y) = f(x) - f(y)$. In particular,

$$(\operatorname{grad}\mathcal{H}(\rho))(x, y) = \log \rho(x) - \log \rho(y) = \log \frac{\pi(y)}{\pi(x)} - \log \frac{p(y)}{p(x)}.$$

Thus, the discrete diffusion "score" is the exponential of the $W_K$-gradient of entropy on the probability simplex (plus a constant term involving the stationary distribution $\pi$ to switch from densities to probabilities). Since, using the logarithmic mean mobility, the discrete heat equation coincides with the $W_K$-gradient flow of entropy (Maas, 2011):

$$D_t \rho_t = -\operatorname{grad}\mathcal{H}(\rho_t). \tag{72}$$

this implies, from a diffusion-modelling perspective, that the standard forward noising process is steepest descent of entropy in the geometry induced by the kernel $K$.

The same viewpoint also yields a principled way to generalize beyond pure heat flow by introducing the potential $V$ to use the full free-energy as a guiding functional:

$$\mathcal{F}(\rho) = \mathcal{V}(\rho) + \beta\,\mathcal{H}(\rho), \qquad \mathcal{V}(\rho) := \sum_{x\in\mathcal{X}} V(x)\rho(x)\pi(x), \qquad \beta > 0. \tag{73}$$

With this functional, the starting distribution can be tuned from the stationary distribution $\pi$ of the chain $K$ to an arbitrary value $p(x) = \frac{\pi(x)e^{-\frac{V(x)}{\beta}}}{Z}$, with $Z$ a rescaling constant. In this sense, the setup introduced in this paper also applies to discrete flow matching (Gat et al., 2024) as a way to describe certain probability flows (still constraining the paths to follow the gradient flow of the functional $\mathcal{F}$) between arbitrary distributions.

Summarizing the quantities involved in our setting and how they relate to modelling choices of discrete diffusion models: $K$ determines which transitions are locally feasible and how they are weighted, $\beta$ controls the strength of the randomness of transitions and can be increased to ensure fast mixing in a generative context, and the potential $V$ steers the dynamics by changing the minimizer of $\mathcal{F}$, hence changing the long-time equilibrium away from the base distribution of the forward heat flow.

Finally, note that some practical discrete diffusion constructions use "absorbing" corruption kernels (Lou et al., 2024), which at first sight may violate the irreducibility and reversibility assumption on the kernel $K$. However, interestingly implementations often regularize in practice these kernels by adding small backward probabilities from the absorbing state into any of the $\mathcal{X}$ states, which improves stability (Lou et al., 2024) and crucially recovers both the irreducibility and reversibility of the regularized absorbing chain. We view this as a positive testament to our theoretical setting, as it is able to justify a prominent experimental practice under the light of precise geometrical properties.

## G. Algorithms

---

**Algorithm 1** Schur–Cholesky solver for the discrete continuity potential (geodesic velocity subroutine)

---

**Input:** $K, \pi, \rho, \dot{\rho}$ with $\langle \dot{\rho}, \mathbf{1} \rangle_\pi = 0$, pinned index $p$
**Output:** $v = \nabla \psi \in T_\rho$
Compute $\Theta(\rho)$ with $\Theta_{ij} = m_\rho(x_i, x_j)$.
$W \leftarrow \mathrm{diag}(\pi)\big(K \odot \Theta(\rho)\big)$.
$M \leftarrow \mathrm{diag}(W\mathbf{1}) - W$.
$A \leftarrow K \odot \Theta(\rho) - \mathrm{diag}\big((K \odot \Theta(\rho))\mathbf{1}\big)$.
$A_g \leftarrow \begin{bmatrix} A \\ e_p^\top \end{bmatrix}, \quad c \leftarrow \begin{bmatrix} -\dot{\rho} \\ 0 \end{bmatrix}$.
Cholesky: $M = L_M L_M^\top$.
Solve $L_M L_M^\top Y = A_g^\top$ for $Y$.
$S \leftarrow A_g Y$.
Cholesky: $S = L_S L_S^\top$.
Solve $L_S L_S^\top \lambda = -2c$.
$\psi \leftarrow -\frac{1}{2} Y \lambda, \quad$ return $v \leftarrow \nabla \psi$.

---

**Algorithm 2** Learning discrete gradient flows from temporal snapshots (training pipeline)

---

**Input:** snapshots $\{\mathcal{Y}_{t_k}\}_{k=0}^T$, $(K, \pi)$, step size $\tau$, batch size $B$, parameters $\theta$
**Output:** trained $\theta$
Estimate $\{\hat{\rho}_{t_k}\}_{k=0}^T$ from $\{\mathcal{Y}_{t_k}\}$ with $\hat{\rho}_{t_k} \in \mathcal{P}_*(\mathcal{X})$.
**for** $k = 1$ **to** $T$ **do**
    Compute $v_k \leftarrow v_{\mathrm{geo}}(\hat{\rho}_{t_k} \to \hat{\rho}_{t_{k-1}})$ using Alg. 1.
**end for**
**repeat**
    Sample $k \in \{1, \dots, T\}$ and minibatch $\{x^{(b)}\}_{b=1}^B \sim \hat{\rho}_{t_k} \odot \pi$.
    $\mathcal{L} \leftarrow 0$.
    **for** $b = 1$ **to** $B$ **do**
        $g \leftarrow \nabla V_\theta(x^{(b)})$, $\beta \leftarrow \beta_\theta$, $s \leftarrow \nabla \log \hat{\rho}_{t_k}(x^{(b)})$, $u \leftarrow v_k(x^{(b)})$.
        $\mathcal{L} \leftarrow \mathcal{L} + \big\| g + \beta s - \frac{1}{\tau} u \big\|_2^2$.
    **end for**
    $\mathcal{L} \leftarrow \mathcal{L}/B$; update $\theta$ with Adam.
**until** convergence or max epochs

---

# H. Graph Classes

**Complete graph**    A complete graph on $n$ vertices is one in which *every* pair of distinct vertices is connected by an edge. Thus each vertex has degree $n - 1$, and the graph is maximally dense. This extreme connectivity makes the complete graph a useful upper bound in diffusion, mixing, and spectral experiments.

**Erdős–Rényi**    In the Erdős–Rényi model $G(n, p)$, each possible edge between the $\binom{n}{2}$ vertex pairs is included independently with probability $p$. The resulting graphs are random and exhibit phase transitions in connectivity, giant components, and degree distributions. This model is a classical baseline in random graph theory: see Erdös & Rényi (1959).

**$d$-Regular**    A $d$-regular graph is one in which every vertex has exactly degree $d$, offering a model with no degree heterogeneity. Random $d$-regular graphs are often well-balanced, locally tree-like, and useful as a homogeneous testbed in structural and spectral studies.

**Watts–Strogatz (Small-World)**    This model builds graphs that interpolate between a regular ring lattice and a random graph via edge rewiring with probability $\beta$. The result often retains high local clustering (like lattices) while achieving short global path lengths, capturing the "small-world" phenomenon seen in many real networks. See Watts & Strogatz (1998).

**Stochastic Block Model**   In the SBM, vertices are partitioned into communities (blocks). Edges are placed between pairs with probabilities depending only on block membership (higher probability within blocks, lower across). This induces modular or community structure and is a canonical model in clustering and network inference. See also Holland et al. (1983).

**Delaunay triangulation graph**   Given a set of points in the plane, the Delaunay triangulation connects points such that no point lies inside the circumcircle of any triangle formed. The result is a planar graph that favors well-shaped triangles and reflects spatial proximity.

**Euclidean Minimum Spanning Tree**   The EMST over a set of points in the plane is the spanning tree (i.e. minimally connected graph) that minimizes total Euclidean edge length. It produces a sparse backbone reflecting the most efficient geographic connections without cycles.

$k$**-Partite**   A $k$-partite graph divides vertices into $k$ disjoint groups (parts). Edges are permitted only across different groups (none within each group). In the complete variant, all cross-part pairs are connected. This structure generalizes bipartite graphs and enforces blockwise connectivity.

**Grid**   In a grid graph nodes are placed in a rectangular array, with edges connecting orthogonal neighbors. Internal nodes typically have degree 4, edges less, reflecting spatial adjacency in 2D.

**Torus**   A torus graph wraps a 2D grid with periodic boundary conditions. Every node is structurally equivalent.

**Apollonian**   An Apollonian network is grown recursively by repeatedly subdividing triangular faces: at each step, pick a triangle, insert a new vertex linked to its three vertices, and replace the triangle with three smaller ones. The result is planar, hierarchical, and often exhibits scale-free degree distributions and small-world distances. See also Andrade Jr et al. (2005).

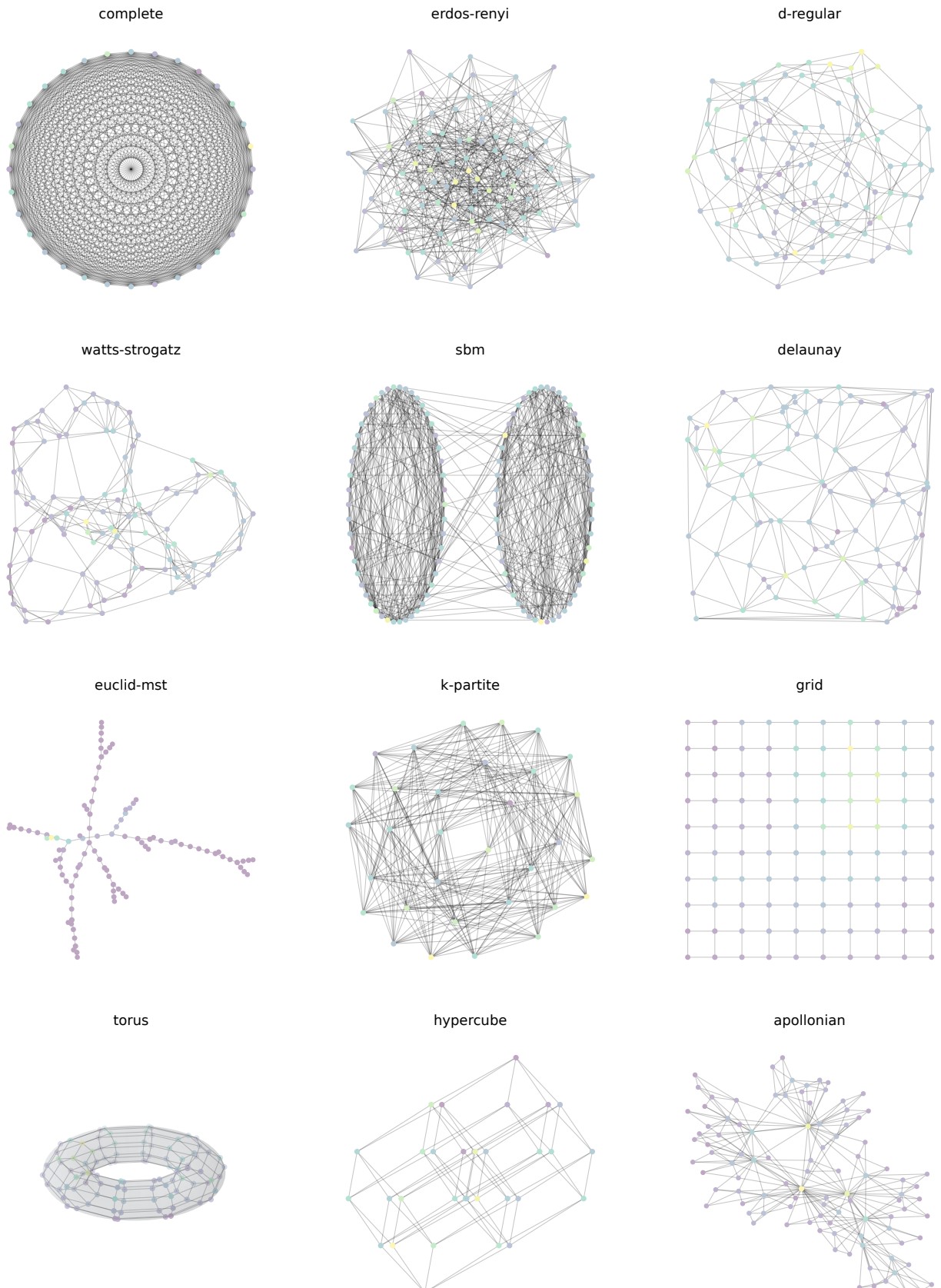

*Figure 5.* Examples of the twelve graph classes we use to benchmark our model.

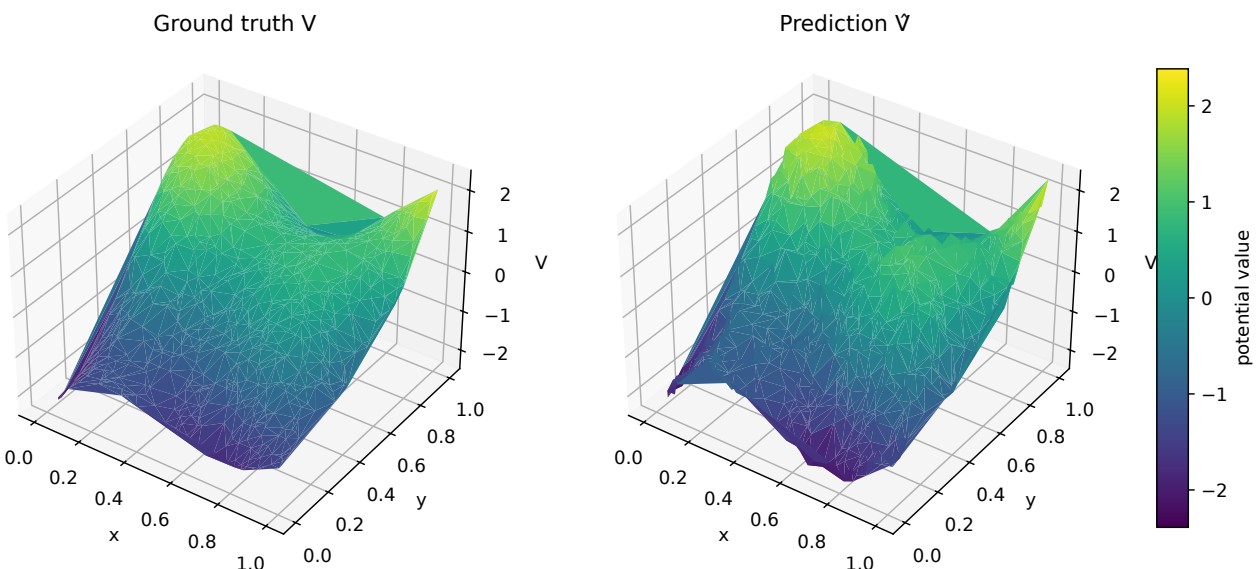

*Figure 6.* A sample smooth potential on a Delaunay graph with 1000 vertices (on the left) and the predicted $V$ from our numerical routine.

