# OpenReview forum: "Learning Discrete Diffusion on Graphs via Free-Energy Gradient Flows"
_ICML.cc/2026/Conference — ICML 2026 regular_

### Official Review · Reviewer_NNGG · 2026-02-14

**Soundness:** 3
**Presentation:** 3
**Significance:** 3
**Originality:** 3
**Overall Recommendation:** 5
**Confidence:** 1

**Summary:**

The paper argues that the standard Wasserstein-2 (W2) gradient-flow/JKO viewpoint breaks on finite discrete spaces, and instead uses a discrete OT-style metric, common discrete diffusion paths (notably the discrete heat equation) can be interpreted as gradient flows of free-energy functionals. The method gives a simple quadatic loss which make training easy.

**Compliance With Llm Reviewing Policy:**

Affirmed.

**Final Justification:**

My concerns have been addressed.

**Key Questions For Authors:**

1. When the number of samples per snapshot is small, is the method still robust?
2. If the forward noising processs is known, could the framework incorporate this?
3. How senstive is the method if K is not reversible or approximated so?

**Limitations:**

yes

**Strengths And Weaknesses:**

I am not quite familiar with the discrete gradient flow. My comments are based on a high-level reading of the paper's claims.

Strength:
1. The learning objective is simple.
2 The computation routine for geodesic estimation is explicit.

Weakness:
1. Evaluation is only on synthetic data
2. The loss is based on first-order necessary conditions. I am not sure about robustness to nonconvexity/spurious stationary points.
3. The geodesic preprocessing has O(N^3), which is unclear how this scales in practice.

---

> ### Author Rebuttal · Authors · 2026-03-31
>
> We thank the reviewer for their thoughtful comments and for recognizing core strengths of the paper: the simplicity of the learning objective and the explicit computational routine for geodesic estimation. We also acknowledge the questions they raise regarding robustness and evaluation, and address them below.
>
> >**Evaluation is only on synthetic data** and **The geodesic preprocessing ... in practice.** and **If the forward ...  incorporate this?**
>
> We answer these jointly due to their related nature. We agree that the current evaluation is restricted to synthetic data, and that the $O(N^3)$ geodesic preprocessing limits the present implementation to small graphs. This fits the goal of the paper, which is to introduce and validate a new snapshot-based learning framework in the discrete $W_K$ geometry. Synthetic graphs are a natural setting in which the underlying free energy is known and recovery can be assessed unambiguously: we also note that the O(N^3) computation is a one-time preprocessing step, not a cost at every optimization step. We anyway agree that scaling of the method is going to be a key aspect of future work, but believe it is currently out of the scope of the work presented here.
> If the forward process were known, the framework could incorporate this information, and this would improve the complexity significantly by relying on popular techniques such as conditional score matching (Song et.al. (2020)) to compute the final loss. We do view the integration of state-of-the-art discrete diffusion machinery as a natural next step for the framework. At the same time, doing so would require additional design choices (such as tuning the expressivity of the network and optimizing specifically for sparse graphs like the Hamming graph) that are orthogonal to the contributions of the present work. We believe such an extension is beyond the scope of the current paper: the main geometric insights remain unchanged, as do the quadratic loss and the geodesic-computing routine introduced here.
>
> >**The loss is based on first-order necessary conditions**
>
> This is an important point: because our strategy imposes first-order necessary conditions, it is in principle sensitive to spurious stationary points. As already noted in the paper, convexity in the $W_K$ geometry is dependent on the graph $K$ and remains, in general, an open problem even for the entropy functional $\mathcal{H}$ (see Maas (2012) and the answer to reviewer jW5H). At the same time, the empirical behavior we observe is consistently stable across a wide range of graph classes, which we view as a nontrivial positive result given the difficulty of understanding convexity in this setting. Our intuition, consistent with the literature, is that more homogeneous, “complete-like” graph structures tend to exhibit stronger convexity and more stable optimization behavior, whereas sparser, more “tree-like” structures are the harder regime.
>
> >**When the number of samples per snapshot is small, is the method still robust?**
>
> We already include an ablation varying the number of samples per snapshot down to 500, where performance remains stable except in the smallest-sample regime; we point the reviewer to Figure 2 in the manuscript as well as Section 6.3.2. To further answer the reviewer’s question, we perform additional experiments with 100 and 200 samples in the same setting: we observe $0.212$ and $0.187$ Hellinger respectively. This slight further degradation is consistent with the structure of the method: our routine is agnostic to the number of samples besides for the estimation of the empirical densities. The degradation as the number of samples drops is hence seemingly due to this density estimation step, and not embedded in the geodesic framework itself.
>
> >**How sensitive is the method if K is not reversible or approximated so?**
>
> The current theory fundamentally assumes reversibility: it is crucially used to obtain symmetry of the geometry. We emphasize that reversibility is a common assumption in many practical settings where CTMC approaches are used. Typical approaches to discrete-space generation use kernels that are irreducible and reversible, which are essential assumptions in the design of the forward process; see, for example, Campbell et.al (2022), Lou et.al. (2023). We also observe an interesting connection with sequence-generation models: common choices of token-level transition matrix are uniform or absorbing toward a special token. Absorbing chains sit at the edge of the reversibility assumption studied here, but practical implementations are slightly regularized from this limit; for example, Lou et al. (2024) use an absorbing base with a small leakage of probability to non-MASK states to avoid pathological behaviors, and this leakage interesting makes the kernels reversible again. We view this as a promising indication that the reversible and irreducible regime captured by our framework is meaningful to design choices made in practice.

---

> > ### Author Rebuttal · Reviewer_NNGG · 2026-04-01
> >
> > I appreciate the efforts from the authors on the replies. I have raised the score and maintained the confidence.

---

> > > ### Author Response · Authors · 2026-04-07
> > >
> > > We thank the reviewer for their attention and positive feedback during what we considered a fruitful review period.

---

### Official Review · Reviewer_C8Sc · 2026-02-24

**Soundness:** 1
**Presentation:** 2
**Significance:** 2
**Originality:** 2
**Overall Recommendation:** 4
**Confidence:** 3

**Summary:**

The goal of this paper is to leverage the theory of probability gradient flow for ML tasks. For this reason, the paper devises a practical algorithm to learn such flows, in particular the functional \mathcal{F}, from temporal snapshots. Finally, the paper performs some experiment on synthetic data to validate the algorithm.

**Compliance With Llm Reviewing Policy:**

Affirmed.

**Final Justification:**

I am okay with the last real-world experiment reported. I have raised my score further.

**Key Questions For Authors:**

My major points are already in the Weakness section.

**Limitations:**

yes

**Strengths And Weaknesses:**

Strengths:

1. A novel perspective on generative modeling involving discrete data with a potential new generative model

2. Some theoretical characterization for the optimization procedure

Weaknesses:

1. Impractical algorithm: To me, this is the most important issue. The proposed framework requires two hard quantities: (1) an accurate density estimator (for high dimensional data), and (2) solving for the quadratic optimization for an exponentially large quantity \mathcal{F}. Both are almost forbidden for practical ML tasks. (Even for a short 8-word sentence, the size of the space N could be as large as ~50000^8.) Note that in contrast, the score function in common discrete diffusion models only scales polynomially (if not linearly) in dimension.

2. Weak experimental setting: Related to the previous point, in the experiment N is chosen to be relatively small (no more than 50). This casts question on the usefulness of the algorithm in practical ML applications.

3. Unclear setting: Throughout the paper, a particular form for \mathcal{F} is chosen (Equation 14), but the motivation is scarce. In particular, it is no clear why a "mirrored" version from the continuous diffusion also makes sense in the discrete case. The intuitive idea of drift plus random fluctuation, while okay for SDEs, does not make sense in CTMC.

4. Unclear novelty: Many results (and proofs in the appendix) are largely drawn from (Maas, 2011).

---

> ### Author Rebuttal · Authors · 2026-03-31
>
> We thank the reviewer for the valuable points raised in the review. We believe that the main disagreement comes from the intended scope of the paper: our goal is not to propose a direct replacement for large-scale sequence-generation pipelines with known noise processes, but to study the inverse problem of learning an unknown functional from temporal snapshots only. We address the reviewer’s concerns point by point below.
>
> >**Impractical algorithm** and **Weak experimental setting**
>
> We answer these concerns jointly due to their related nature. The reviewer is correct that the current implementation is not aimed at large spaces such as language-scale sequence generation, and that our experiments are thus restricted to moderate graph sizes. However, we believe **the comparison is partially against a different problem setting**: in discrete diffusion models for sequence generation, the forward noising process is known, which enables techniques such as conditional score matching (Song et.al (2020)) to be used to simplify the problem; in addition, the graph is typically the sparse Hamming graph, whose structure is itself a crucial inductive bias for scalability (see, e.g., Lou et al., 2024).
>
> **Our framework does not assume access to the ground-truth forward process, and instead works only from temporal snapshots of an unknown evolution.** This introduces a computational tradeoff: unlike classical discrete generative models, **we cannot leverage the reductions that make the joint score loss tractable**. Instead, we directly estimate the densities $\rho_t$ from the observed temporal snapshots (which is tractable at our scale) and then recover the joint scores via our quadratic optimization problem. We do see the integration of the computational machinery of state-of-the-art discrete diffusion models as a prominent next step for our framework; however, doing so would require a number of additional design choices, such as tuning network expressivity and optimizing specifically for sparse graphs like the Hamming one, that are orthogonal to the theoretical and practical contributions of the present paper. We will clarify this connection in the updated manuscript, but believe this extension is beyond the scope of the current work.
>
> >**Unclear novelty**
>
> We agree that the underlying $W_K$ geometry builds on prior mathematical work, as stated in the paper. Our claim of novelty is not the introduction of the geometry itself, but **showing how the related theory can be turned into a computational framework for machine learning**. Concretely, the novel components we introduced to this end are:
> - the implementation-critical Theorem 4.1, stating that the discrete free-energy JKO problem has a unique minimizer in the interior of the simplex for any $\tau>0$, which is essential because the $W_K$-Riemannian structure is only available on the interior;
>  - the geodesic-computing routine together with Proposition C.3 identifying geodesics as the gradient of $W_K^2$;
>  - the resulting quadratic training objective and empirical validation for learning from temporal snapshots, which constitute the first discrete adaptation of the first-order learning strategy popular in the continuous case;
>
> We also consider it an independent merit of this work **to bring discrete gradient-flow tools to the attention of the machine learning community**, where they have so far received little attention.
>
> >**Unclear setting**
>
> We thank the reviewer for pointing out that the motivation for our choice of functional was under-explained. We think the free-energy functionals to be a prominent candidate for a number of reasons. First, the entropy functional is already a natural candidate in our setting: in the geometry induced by $W_K$, it is the functional associated with the popular discrete heat equation. Adding a potential is then the simplest principled way to make the dynamics more expressive: this already allows probability curves and equilibria to move beyond pure relaxation to the stationary distribution of the chain and, as discussed in Appendix F, to connect much more general pairs of distributions.
>
> Furthermore, we had no intention to motivate our choice via a loose mirroring of the SDE setting, but to point out that free-energy dynamics naturally model discrete stochastic systems in which random fluctuations coexist with an external field or potential shaping trajectories. Similar features appear in physics and chemistry settings such as in Ising models in an external field (Van Hao et.al (2019)) and chemical reaction networks (Mielke (2011)). We believe this provides a meaningful motivation for studying free-energy functionals in the discrete setting.
>
>
> We hope we have addressed all of the reviewer’s concerns on the paper, and are happy to respond to any further questions they might have.

---

> > ### Author Rebuttal · Reviewer_C8Sc · 2026-04-01
> >
> > My main concern still stands, that even in a different problem setting and even where there is no baseline methods to compare against, the proposed method suffers from the need of an accurate density estimator and the computation power to solve for an exponentially large quadratic optimization problem. Moreover, since this is a built-in issue of the proposed method, I do not believe it can be easily addressed in such a short time.
> >
> > Could the authors identify any real-world setting with a similar size of the state space as in the paper that is interesting to the majority of the ML community? If so, I might change my mind.
> >
> > That said, I think there is still some novelty on the modeling side, so I have raised my score slightly.

---

> > > ### Author Response · Authors · 2026-04-07
> > >
> > > We thank the reviewer for their constructive feedback.
> > > Motivated by this, we looked into a new experiment using real-world data to strengthen our experimental evaluation. We focused on single-cell trajectory inference, particularly for predicting population-level cell-type counts. In this application, temporal snapshots are often aggregated into a small number of clusters or cell types (typically between 15 and 40) with an expert-provided lineage graph (Klein et.al, 2023, Lange et.al 2024). As a proof of concept, we train a time-varying version of our model on the mouse gastrulation atlas (Pijuan-Sala et al., 2019), a benchmark dataset in the trajectory inference literature containing ~116,000 cells across 8 timepoints with 34 expert-annotated cell types and a known developmental lineage graph. We report average Hellinger numbers against the direct Markov baseline requested by reviewer jW5H.
> > >
> > > | Method | Hellinger ↓ |
> > > |---|---:|
> > > | Direct Markov Baseline | 0.36 |
> > > | Ours | 0.06 |
> > >
> > > We think this result is promising, as it directly translates our setup to a non-synthetic dataset in a prominent optimal transport application domain (Tong, Fatras, Malkin et al. 2024; Bunne et al. 2023; Terpin et al. 2024). Existing approaches make the problem continuous by converting gene counts into PCA space, while operating directly in discrete gene space has remained out of reach. We provide an initial yet theoretically sound computational framework, though it is currently limited to coarse cell-type features and not yet applicable directly to genes. Scaling remains the most impactful next improvement for our approach, as noted by this reviewer and stated in the paper. We are happy to strengthen the related discussion if the reviewer feels it’s necessary.
> > >
> > > We thank the reviewer for the stimulating questions and feedback they provided in what was, on our end, a fruitful discussion period.

---

### Official Review · Reviewer_k44i · 2026-03-11

**Soundness:** 4
**Presentation:** 3
**Significance:** 3
**Originality:** 2
**Overall Recommendation:** 5
**Confidence:** 4

**Summary:**

In this paper, the authors proposed a framework to solve the following problem: if we can observe the diffusion evolution (discrete trajectory) of a particle, what is the energy function that drives this evolution? In simple language, in Euclidean space, suppose that we can observe a trajectory, what is the potential function $f$ that has that trajectory as its gradient flow? The authors studied this problem for flows defined in the space of distributions $\mathcal{P}(X)$ (flow = diffusion process) whose state space $X$ is discrete. If the state space is continuous, this problem has been studied intensively in the literature via the JKO framework. When the state space is discrete, the JKO operator with the Wassertein-2 distance cannot be used, and it is known that the Wasserstein-2 metric in the JKO operator should be replaced by the W_K metric defined based on some appropriate mobility [Maas, 2011], where K is the Markov kernel of the probability evolution. The main contribution of the paper is to propose a practical method to learn the energy based on the newly defined JKO operator.

**Compliance With Llm Reviewing Policy:**

Affirmed.

**Final Justification:**

After considering the author's rebuttal and another careful look at the paper, I increased my score from 4 to 5.

Overall, I believe this paper has some non-trivial contributions. Although the logic of learning diffusion by the JKO scheme is established in the literature, doing so in a graph requires much adaptation. Still, as Reviewer C8Sc pointed out, some practical scalability concerns may still remain.

**Key Questions For Authors:**

- Do you need to learn the Markov kernel K?
- Why do you need to write probability densities w.r.t. $\pi$? Can we just use the counting measure as the underlying measure for the space?

**Limitations:**

yes

**Strengths And Weaknesses:**

# Strenth
- The writing is easy to follow with sufficient background presented.
- The method has practical merit as it is fast, accurate, and bridges the gap between pure math theory and computational ML.

# Weakness
- As the authors also acknowledge, most of the theory is known. The contribution is on how to efficiently compute geodesics in this new $W_K$ geometry. The quadratic loss based on the first-order optimality of the JKO is also not new, see: Terpin, A., Lanzetti, N., Gadea, M., & Dörfler, F. (2024). Learning diffusion at lightspeed. Advances in Neural Information Processing Systems, 37, 6797-6832.
- The framework is restricted to learn the energy of the form of the sum of entropy and potential. In practice, energy can take other forms, e.g., interaction energy.

---

> ### Author Rebuttal · Authors · 2026-03-31
>
> We thank the reviewer for their insightful comments and questions about the paper. We are pleased that they appreciated the quality of the exposition and recognized its merits as both a fast and accurate solution as well as a theoretically grounded one. We also acknowledge the questions they pose: we address them one by one in what follows.
>
> >**Do you need to learn the Markov kernel $K$?**
>
> In this work, the Markov kernel $K$ is assumed to be a known quantity, and it models along which states is the probability allowed to flow at a given timestep. Notice that, in practice, situations where $K$ is known are common; in discrete generative models the underlying graph is fixed (typically the Hamming graph, see Appendix F in the manuscript), and the dynamics follow the relative entropy gradient flow w.r.t. its stationary distribution. In other applied settings the graph encodes known structure about the problem where the geometry is prescribed, for example chemical reaction networks (see Mielke (2011)). We remark that this assumption is only needed to generally constrain the geometry of the studied discrete probability flow to align with the presented theory: the exact quantitative flow is learned through the functional $\mathcal{F}$.
>
> In general, we see the problem of instead predicting the graph $K$ given the same trajectories (perhaps with some constraint on the noise process) as an interesting orthogonal counterpart to the one presented here, which would merit an entirely different publication. Our framework can be combined with standard graph-learning ideas for parametrizing $K$ (for example Kalofolias, V. (2016) and Dong et.al. (2016)) and either jointly fit $K$ and the functional or assume a known process like the heat equation.
>
> >**Why do you need to write probability densities w.r.t. $\pi$? Can we just use the counting measure as the underlying measure for the space?**
>
> Yes, we can parametrize everything either in the language of densities or in that of probabilities: given that we work in the setting where $K$ is available, computing $\pi$ from $K$ is straightforward, and then one can use $\pi$ to switch between probabilities and densities with no significant computational cost. We chose to present our theory in the language of densities to be more adherent to the original setting of Maas (2011) which is formulated in terms of those.
>
> >**The quadratic loss based on the first-order optimality of the JKO is also not new**
>
> We believe this is a misrepresentation of our contribution, and think the reviewer might have missed a critical point of the work. Learning functionals from first-order optimality conditions is not new in the continuous setting. *Our claim of novelty is its first integration into a discrete space learning framework*. In particular, turning the discrete theory into a learning method **requires ingredients that are not provided by the continuous versions**:
>
> - a discrete JKO formulation adapted to snapshot-based learning, which we distill from a decade-old paper in mathematics which has no implications known in machine learning
> - Theorem 4.1 guaranteeing that the free-energy JKO minimizer exists, is unique, and lies in the interior of the simplex, this is a novel result, crucially enabling learning in this context
> -  a practical geodesic-based preprocessing/training pipeline, which we derive from these first-principles foundations.
>
> Without these elements, **the application of functional learning techniques which originate from the continuous setting** (such as the first-order necessary conditions of Terpin et.el (2024) or the bilevel optimization of Bunne et.al (2021)) **to discrete domains would be impossible**. In fact, this structural impossibility is explicitly stated in Lemma 2.1.
>
> >**The framework is restricted to learn the energy of the form of the sum of entropy and potential. In practice, energy can take other forms, e.g., interaction energy.**
>
> We agree that the present framework focuses on the free-energy form, and does not yet cover more general energies such as interaction terms. As the entropy term is the canonical functional associated with the discrete heat equation, the addition of a potential yields the natural class that mirrors widely used functionals like the ELBO while allowing the flow to connect almost any pair of distributions. We view this as the right starting point for a first computational framework. Extending the method to richer energies, including interaction terms, is an interesting next step: it sounds straightforward for a practitioner which expects interactions to be present in their problem to accommodate them by proving a correspondent version of Lemma C2/C4/Proposition C3, and using a third dedicated network head to predict the new term. We thank the reviewer for pointing us to this generalization.

---

> > ### Author Rebuttal · Reviewer_k44i · 2026-04-02
> >
> > Thanks for addressing my concerns/questions. Overall, I maintain a positive rating on the work.

---

> > > ### Author Response · Authors · 2026-04-07
> > >
> > > We thank the reviewer for their attention and positive opinion on our paper. We also agree that scaling the graph dimension, as noted by this and other reviewers, is the most impactful next challenge for this approach, which we plan on tackling by incorporating further inductive biases in domains where the ground-truth noising process is known (such as discrete-space generative modeling, see response to Reviewer NNGG). We are pleased that the reviewer appreciated the methodological and theoretical contributions of this work and that they agree on them being of independent interest to the machine learning community.

---

### Official Review · Reviewer_jW5H · 2026-03-11

**Soundness:** 3
**Presentation:** 3
**Significance:** 3
**Originality:** 3
**Overall Recommendation:** 5
**Confidence:** 3

**Summary:**

This paper develops a discrete analogue of the JKO based learning approach for gradient flows on probability spaces. The standard $W_2$ framework fails on finite state spaces. The authors work instead with the $W_K$ metric, under which the discrete heat equation is the gradient flow of the relative entropy. They prove existence and uniqueness of the discrete JKO minimizer and derive first order optimality conditions. They observe that these conditions are linear in the unknown potential and noise parameter given precomputed geodesic velocities. This reduces training to a quadratic regression problem. Geodesic velocities are obtained as a preprocessing step and the method only requires temporal snapshots of the dynamics. Experiments cover 10 synthetic graph classes.

**Compliance With Llm Reviewing Policy:**

Affirmed.

**Final Justification:**

The paper presents a clean theoretical framework and the rebuttal was thorough. The forward simulation baseline confirms that the $W_K$ geometry adds value beyond naive density matching. The $\tau$-sensitivity discussion honestly acknowledges the open questions around geodesic convexity.

My main appreciation is for the theoretical contributions.  The translation of the Maas framework into a working algorithm, the linearity of the first-order conditions in the unknowns and the connection to discrete scores. The theory is sound and the presentation is clear.

My remaining reservations are the lack of larger experimental evidence.  That said, I think the contributions are very much sufficient for acceptance.

**Key Questions For Authors:**

1.  At inference, $\nabla V$ and $\beta$ define a rate matrix $Q$ and densities are propagated via matrix exponential (Appendix D). Could one use this same forward procedure as the training objective, propagating densities given $(\nabla V, \beta)$ and minimizing the mismatch with observed snapshots? Could this serve as a baseline?

2. How sensitive is choice of $\tau$, in continuous $W_2$ JKO the discretization error is controlled, is there an analogous estimate here?

**Limitations:**

yes

**Strengths And Weaknesses:**

**Strengths**
- Natural extension of the $W_2$ JKO learning framework to finite state spaces. The choice of $W_K$ is well motivated by the identification of the heat equation with the gradient flow of entropy.
- Thm 4.1 establishes that the JKO minimizer is unique and strictly interior for any $\tau > 0$.
- The first-order conditions (Thm 4.3) decompose the gradient into $\nabla V + \beta \nabla \log \rho - v_{\text{geo}}/\tau = 0$, which is linear in the unknowns given the geodesic velocity.
- The previous observation reduces training to regression. The separation of geometric preprocessing from learning is clean.
- Appendix F connecting the framework to CTMC based discrete diffusion is a nice addition. The identification $\text{grad} H(\rho) = \nabla \log \rho$ provides a geometric interpretation of the discrete score used in CTMC based diffusion models.

**Weaknesses**

- Evaluation is Hellinger distance on predicted densities only. Since the network directly outputs $\nabla V$, reporting $L^2$ error on $\nabla V$ and (relative) error on $\beta$ would strengthen the quantitative evaluation.
- All experiments are on synthetic graphs with at most 50 nodes. A real data experiment (at small scale) would strengthen the practical motivation.
- There is only one baseline, which if I understand correctly is zero shot across graphs.

---

> ### Author Rebuttal · Authors · 2026-03-31
>
> We thank the reviewer for their insightful comments. We are pleased that they appreciated several aspects of the paper, such as the clean regression form the setup allows and the theoretical connection with generative setups. We also acknowledge the weaknesses and questions they raised, and address them in turn below.
>
> >**Evaluation is Hellinger only** and **There is only one baseline** and **At inference... Could this serve as a baseline?**
>
> We answer these jointly due to their related nature. The strategy detailed by the reviewer is indeed a valid baseline. We implemented a version of it and additionally reported relative $L^2$ error on $\nabla V$ and  $\beta$ for both methods. The baseline works in density space with the same estimated $\rho_t$ as our method: at each observed snapshot $\hat\rho_t$, it computes the rate matrix $Q_t$, backpropagates through $\exp(\Delta t  \cdot Q_t^T)$, and minimizes squared Hellinger distance directly via Adam on $\nabla V$ and $\beta$. We tested this on one sample for each graph class with graph size 30 , 5k samples $\beta \in \{0.01,0.1,0.2\}$. We report numbers aggregated over all graph classes.
>
> |  | Hellinger  | Rel. error $\nabla V$ | Rel. error $\beta$ |
> |---------|------|------|--------|
> | Ours | 0.152 | 0.103 | 0.376 |
> | Baseline| 0.182 | 1.873 | 1.008 |
>
> Despite not optimizing Hellinger directly, our method is slightly better on Hellinger and significantly better at recovering the underlying geometry. The direct baseline appears to learn a spurious solution of the dynamics that fits Hellinger reasonably well but fails to identify the ground-truth $V$ and $\beta$. The discrepancy between the errors on $\nabla V$ and $\beta$ is also reasonable: with $V$ sampled in the $[-1,1]$ range, the effect of $V$ on the dynamics is much stronger than that of $\beta$. As a sanity check, we test our method in the setting $V \equiv 0$ also, where it recovers $\beta$ almost perfectly with an average error of $0.03$.
>
> We agree that a real-world use case would be the ideal final component of the evaluation pipeline; however, we think at this stage the synthetic setting is particularly appropriate given the known and tunable nature of the underlying geometry. Exploring real-world applications is one of the main next steps we see for this setup.
>
> >**How sensitive is choice of $\tau$ , in continuous $W_2$ JKO the discretization error is controlled, is there an analogous estimate here?**
>
> This is an interesting question, and we studied it in more detail after reading the review. In our understanding, the situation is more delicate than in the continuous case: for some graphs $K$ one can expect the same $\mathcal O(\tau^{1/2})$ bound as in the continuous JKO setting, while for general $K$ this remains open.
>
> In the continuous case, the classical route (e.g. Santambrogio, Sec. 8.3; Ambrosio-Gigli-Savaré, Sec. 11.1.3) is to assume $\lambda$-geodesic convexity in $W_2$, derive a discrete energy-dissipation / variational inequality for the JKO minimizers, and then pass to the continuous limit to get the standard $\mathcal O(\tau^{1/2})$ bound.
>
> In the discrete case, the same strategy would work provided the free-energy functional enjoys some kind of geodesic convexity in $W_K$. Unfortunately, the geodesic convexity of $\mathcal H$ in $W_K$ is currently an open  problem for a general graph $K$. We point the reviewer to Erbar and Maas (2012) and Erbar et al. (2019) for partial results, where a dependence on the graph $K$ emerges, with more “complete-graph”-like structures showing stronger convexity. For such graphs, the classical AGS strategy seems fully replicable. It is also possible that a different proof exploiting the finite dimensionality of our setting could work; our Theorem 4.1 is encouraging in this direction, since for free energies the JKO scheme stays away from the degenerate boundary of $\mathcal P(\mathcal X)$ for every $\tau$. While we do not have a proposed proof strategy at this stage, we are excited to study this problem in the future.

---

> > ### Author Rebuttal · Reviewer_jW5H · 2026-04-04
> >
> > Thanks for the detailed rebuttal. The forward simulation baseline is a nice to see and the discussion of $\tau$-sensitivity was helpful. However I think my score is adequate and I will maintain it.

---

> > > ### Author Response · Authors · 2026-04-07
> > >
> > > We thank the reviewer for their appreciation of our paper and for their interesting questions which helped us strengthen our results. We are pleased they valued the theoretical and methodological contribution of the work, and are grateful for what we consider a fruitful review period.

---

### Decision · Program_Chairs · 2026-04-30

**Decision:**

Accept (regular)

**Comment:**

The authors propose a novel learning framework for learning discrete diffusion processes on graphs by extending the JKO gradient flow to discrete spaces using the $W_K$ metric. The resulting method leverages first-order optimality conditions to recover the underlying free-energy functional from temporal snapshots. Reviewers agree that the authors provide a clear and principled bridge between discrete diffusion modeling and discrete gradient flow with metric $W_K$, and turn the theoretical framework into a practical learning approach. However, the reviewers also raised concerns on its scalability limitation.

Overall, we think the submission provides novel perspective on discrete diffusion, conceptually elegant despite limitations in its scalability. Therefore, we recommend acceptance.